# Sharper Convergence Rates for Nonconvex Optimisation via Reduction Mappings

**Evan Markou**
Australian National University
evan.markou@anu.edu.au

**Thalaiyasingam Ajanthan**
Australian National University & Pluralis Research
thalaiyasingam.ajanthan@anu.edu.au

**Stephen Gould**
Australian National University
stephen.gould@anu.edu.au

## Abstract

Many high-dimensional optimisation problems exhibit rich geometric structures in their set of minimisers, often forming smooth manifolds due to over-parametrisation or symmetries. When this structure is known, at least locally, it can be exploited through reduction mappings that reparametrise part of the parameter space to lie on the solution manifold. These reductions naturally arise from inner optimisation problems and effectively remove redundant directions, yielding a lower-dimensional objective. In this work, we introduce a general framework to understand how such reductions influence the optimisation landscape. We show that well-designed reduction mappings improve curvature properties of the objective, leading to better-conditioned problems and theoretically faster convergence for gradient-based methods. Our analysis unifies a range of scenarios where structural information at optimality is leveraged to accelerate convergence, offering a principled explanation for the empirical gains observed in such optimisation algorithms.

## 1 Introduction

First-order gradient methods are the workhorse for large-scale optimisation in machine learning and data science due to their simplicity and scalability. However, the objective functions encountered in these settings often exhibit highly non-convex and intricate loss landscapes, stemming from various factors such as over-parametrisation, compositions of nonlinear functions, and underlying data distributions, to name a few [1–4]. Despite this complexity, gradient-based methods perform remarkably well in practice, achieving local linear convergence under mild regularity conditions—such as the widely studied Polyak-Łojasiewicz (PŁ) condition [5–9]. This apparent tension between theoretical difficulty and empirical success motivates a deeper understanding of the geometry of loss landscapes and its role in shaping optimisation dynamics.

Let us consider the geometry of the solution spaces, which are shown to have manifold-like structures, rather than isolated points for many machine learning problems due to over-parametrisation, symmetries, or latent invariances [10, 6]. Such structured sets of minimisers are not merely theoretical curiosities—they naturally arise in a variety of real-world applications. For instance, in deep neural networks, over-parametrisation often leads to entire manifolds of local minima [10] related by symmetries [11–13]. Furthermore, recent findings in neural collapse [14] reveal that optimal solutions often exhibit highly regular, symmetric, low-dimensional structures across layers [15–20]. In matrix factorisation problems [21] such as dictionary learning [22–25], low-rank matrix completion [26, 27], and tensor decomposition problems [28, 29], solutions are only identifiable up to scaling or orthogonal

39th Conference on Neural Information Processing Systems (NeurIPS 2025).

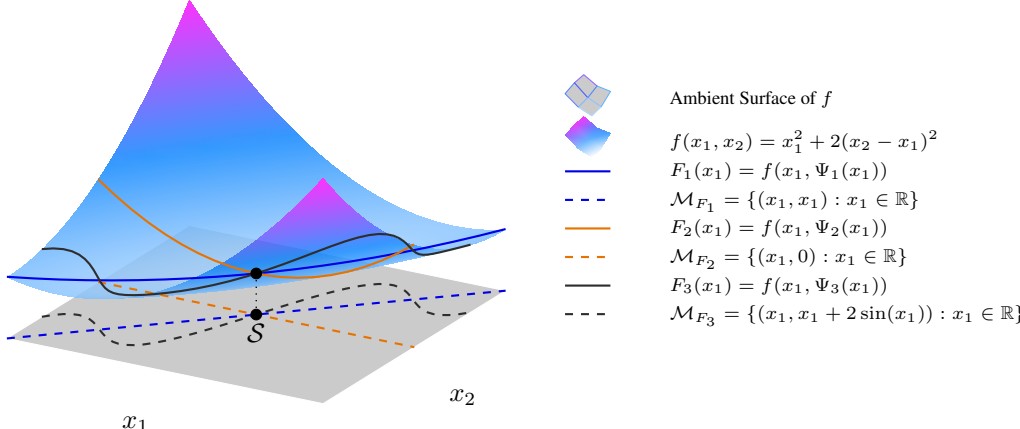

Figure 1: Illustration of how well-designed reduction mappings iron out worst-case curvature. The opaque surface depicts the graph of the function $f : \mathbb{R}^2 \to \mathbb{R}$, lifted above the ambient domain (grey plane) for visualisation purposes. The function has a single global minimum $\mathcal{S} = \{(0,0)\}$. In general, $\mathcal{S}$ can be a set of non-isolated points. The blue curve shows the restriction of $f$ along the mapping $\Psi_1 : x_1 \mapsto x_1$, where the high-curvature quadratic component cancels, yielding a flatter profile in $x_1$. In contrast, the orange curve corresponds to the mapping $\Psi_2 : x_1 \mapsto 0$, which preserves most of the steep curvature of $f$. The grey-green curve traces the restriction along a nonlinear sinusoidal mapping—an example of a poorly designed reduction, which introduces additional curvature into the problem. Dashed curves on the ambient domain represent the images of these mappings as one-dimensional submanifolds $\mathcal{M}$. The submanifolds have a non-empty intersection with $\mathcal{S}$.

transformations [30], reflecting the inherent symmetries of these models. Similar invariances exist in problems such as phase retrieval [31–34] and blind deconvolution [35–40]. These symmetries induce structured sets of minimisers, often forming smooth manifolds or discrete equivalence classes, reflecting invariance under these transformations [30]. These examples all share a common theme: *the objective exhibits invariances that endow the set of solutions with rich geometric structure.*

In this work, we study how the geometric structure of the solution set can be systematically exploited to accelerate convergence. We focus on reduction mappings—reparametrisations that encode known components of the solution manifold, and study their effect on the local curvature. In practice, these reductions often arise from inner optimisation problems, thereby reformulating the original problem into a bi-level optimisation problem and yielding a lower-dimensional objective. While intuitively promising, not all such reduction mappings are beneficial. Even if optimal in value, a poorly designed reduction can distort local curvature and hinder convergence. We develop a rigorous framework to identify when and how these mappings lead to provable improvements in outer iteration complexity for gradient descent.

To build geometric intuition, Figure 1 illustrates how different reduction mappings affect the local curvature of the objective along their respective subspaces. For more details on this example, refer to Appendix A. While some mappings reveal a well-conditioned, flattened profile, others retain or even exaggerate steep curvature. This highlights a central idea of our work: *the convergence behaviour of gradient methods depends not just on the presence of structure, but on how effectively it is incorporated into the optimisation process.*

## 1.1 Contributions

We provide a systematic characterisation of when reduction mappings lead to provable gains in optimisation efficiency. Specifically, we identify conditions under which the reduced objective exhibits a strictly smaller smoothness constant and a strictly larger *sharpness* constant[1]. Together, these improvements yield a strictly better condition number, leading to faster worst-case convergence

---

[1]A general term encompassing Polyak-Łojasiewicz (PŁ), quadratic growth (QG), and related properties.

rates for gradient-based methods applied to the reduced problem. The results hold under standard regularity assumptions and apply to both affine and nonlinear mappings. Specifically we contribute the following:

- Theorem 1 shows that for affine reduction mappings, the smoothness constant of the reduced objective is strictly smaller than that of the full objective.
- Theorem 2 extends this result to general nonlinear mappings, establishing improved smoothness under mild regularity assumptions.
- Theorem 3 demonstrates that the sharpness constant of the reduced problem is strictly larger, implying stronger curvature near the minimisers.
- Corollary 3 combines the above to show that the condition number of the reduced problem is strictly better, leading to faster convergence of gradient descent under the PŁ condition.

For completeness, we include a number of supporting lemmas and additional theoretical results in the appendices.

## 1.2 Related Work

Reparametrisations and geometry-aware optimisation have long been used to exploit structure and improve conditioning in nonconvex problems [41]. Examples include normalisation methods [42, 43], which can be interpreted as explicit reduction mappings improving feature geometry and convergence [44]; neural collapse [14] formulations, where our framework captures both fixed classifier parametrisations [45] and dynamic equiangular tight frame (ETF) projections [46]; and gauge-fixing strategies in problems with symmetries, offering a lightweight alternative to quotient manifold optimisation [47, 48]. Preconditioned methods also relate closely to our approach, with connections to natural gradient descent [49, 50], adaptive preconditioning [51, 52], and studies of parameter space geometry under reparametrisation [53]. Our framework generalises these by treating reduction mappings as intrinsic geometry-aware preconditioners, unifying and extending prior work on preconditioning and geometry adaptation. Finally, classical variable elimination and bilevel optimisation methods [54, 55] can be seen as special cases of reduction mappings, where our analysis goes beyond dimensionality reduction to provide precise improvements in conditioning and convergence rates. A more detailed discussion is provided in Appendix B.

## 2 Setup, Notation, and Preliminaries

We consider general unconstrained optimisation problems of the form

$$\underset{x \in \mathbb{R}^n}{\text{minimise}} \, f(x) \,, \tag{1}$$

where $f : \mathbb{R}^n \to \mathbb{R}$ is a $C^2$ (twice differentiable), possibly non-convex function. We assume that the set of minimisers of $f$ is not discrete, but instead forms a non-isolated set. Specifically, we define the set of all local minima in a neighbourhood as

$$\mathcal{S} = \{x \in \mathbb{R}^n : x \text{ is a local minimum of } f \text{ and } f(x) = c\} \,. \tag{2}$$

Without loss of generality, we assume the minimum of $f$ is zero, *i.e.*, $c = 0$. We further assume that, locally around any minimiser, the function $f$ satisfies the PŁ condition with constant $\mu > 0$ ($\mu$-PŁ), that is,

$$f(x) \leq \frac{1}{2\mu} \|\nabla f(x)\|^2 \,. \tag{PŁ}$$

Other related conditions commonly imposed in non-convex optimisation include the quadratic growth (QG) condition [56] and the error bound (EB) condition [57]. These can be defined as follows, where dist is the classical Euclidean distance: $f$ satisfies the quadratic growth condition with $\mu > 0$ around a minimum if

$$f(x) \geq \frac{\mu}{2} \text{dist}^2(x, \mathcal{S}) \,. \tag{QG}$$

Also, $f$ satisfies the error bound condition with $\mu > 0$ around a minimum if

$$\mu \, \text{dist}(x, \mathcal{S}) \leq \|\nabla f(x)\| \,. \tag{EB}$$

Numerous works derive convergence rates for gradient-based methods under these assumptions [58–61]. Another important condition is the Morse–Bott property [62–64], which generalises the classical Morse theory framework [65]. While Morse functions require all critical points to be isolated and non-degenerate, the Morse–Bott condition relaxes this by allowing the set of critical points to form smooth manifolds, as long as the Hessian is non-degenerate in directions normal to these manifolds [62]. This setting is particularly relevant in optimisation problems where symmetries or invariances naturally give rise to non-isolated minimisers lying on structured sets and hence singular Hessians.

**Definition 1 (Morse–Bott Property)** *Let $\bar{x}$ be a local minimiser of $f$ with associated set of minimisers $\mathcal{S}$ and $\mathrm{T}_{\bar{x}}\mathcal{S}$ be the tangent space of $\mathcal{S}$ at $\bar{x}$. Then $f$ satisfies the Morse–Bott property at $\bar{x}$ if*

$$\mathcal{S} \text{ is a } C^1 \text{ submanifold around } \bar{x} \qquad \text{and} \qquad \ker \nabla^2 f(\bar{x}) = \mathrm{T}_{\bar{x}}\mathcal{S}\,. \tag{MB}$$

*Furthermore, if there exists a uniform $\mu > 0$ such that for all vectors $v$ normal to $\mathcal{S}$ one has*

$$\langle v, \nabla^2 f(\bar{x})\, v \rangle \geq \mu \, \|v\|^2\,,$$

*then we say that $f$ satisfies the $\mu$-MB property.*

Notably, for $C^2$ functions, it has been shown [64] that these conditions are essentially equivalent—up to potential degradations in the constants or reductions in the neighbourhoods where they hold. Therefore, throughout this work, assuming any one of these conditions allows us to invoke the others interchangeably. Under (MB), we refer to set of minimisers $\mathcal{S}$ as the *solution manifold*.

**Assumption 1 (Standing Assumptions on Function $f$)** *Suppose there exist constants $L, \beta, \mu > 0$ and a compact neighbourhood $\mathcal{N} \subset \mathbb{R}^n$ around a minimiser $\bar{x}$ such that the following hold:*

1. *(**Interpolation**) The function $f$ attains its infimum at zero, and the set of minimisers $\mathcal{S}$ is non-empty.*

2. *(**Smoothness**) The function $f$ is $L$-Lipschitz continuous and $\beta$-smooth on $\mathcal{N}$; that is, its gradient is $\beta$-Lipschitz continuous on $\mathcal{N}$.*

3. *(**PŁ condition**) The function $f$ satisfies the (PŁ) condition at every point in $\mathcal{N}$.*

## 2.1 Reparametrisation via Reduction Mappings

We begin by decomposing the variable $x$ of the function $f$ into two components, $x_1 \in \mathbb{R}^{n_1}$ and $x_2 \in \mathbb{R}^{n_2}$, such that $n = n_1 + n_2$, with $n_1 < n$. Whenever the problem structure or prior knowledge permits, we assume that the component $x_2$ exhibits a known geometric structure at optimality. This allows us to introduce an appropriate mapping—typically a projection or an implicit parametrisation—that explicitly encodes this structure by setting $x_2$ as a function of $x_1$. This leads to a reduction mapping, where $x_2$ is fixed to lie on its optimal structure, enabling us to reformulate the problem by optimising only over the remaining variables $x_1$.

**Definition 2 (Reduction Mapping and Reduced Function)** *Let $\Psi : \mathbb{R}^{n_1} \to \mathbb{R}^{n_2}$ be a $C^2$ mapping representing the known geometric structure of $x_2$ at optimality. We define the* reduction mapping *$\Phi : \mathbb{R}^{n_1} \to \mathbb{R}^n$ as*

$$\Phi(x_1) := \big(x_1, \Psi(x_1)\big)\,. \tag{3}$$

*We then define the reduced objective $F : \mathbb{R}^{n_1} \to \mathbb{R}$ as the pullback of $f$ along $\Phi$ as*

$$F(x_1) := f\big(\Phi(x_1)\big) = f\big(x_1, \Psi(x_1)\big)\,. \tag{4}$$

*Since $f$ and $\Phi$ are $C^2$, the reduced function $F$ is also $C^2$.*

**Definition 3 (Graph Manifold)** *We define the graph manifold of the reduction mapping $\Phi$ as*

$$\mathcal{M}_F := \{\Phi(x_1) = \big(x_1, \Psi(x_1)\big) \mid x_1 \in \mathbb{R}^{n_1}\}\,. \tag{5}$$

*Assuming $\Psi$ is $C^2$, the manifold $\mathcal{M}_F$, which we will refer to as the* feasible manifold*, is a globally embedded $C^2$ submanifold of $\mathbb{R}^n$ of dimension $n_1$, since $\Phi$ is globally injective and $C^2$.*

**Assumption 2 (Standing Assumptions on the Reduced Function $F$)** *Assume the following hold within a compact neighbourhood $\mathcal{N} \subset \mathbb{R}^n$ defined around a minimiser $\bar{x}$:*

1. *The reduced objective $F$ is $\beta_F$-smooth on $\mathcal{N}$.*

2. *The intersection $\mathcal{M}_F \cap \mathcal{S}$ is non-empty. We denote this set as $\mathcal{S}_F := \mathcal{M}_F \cap \mathcal{S}$, which forms the set of minimisers of $F$.*

**Notation**  *In the sequel, we use the subscript $f$, to denote all constants associated with the original objective $f$ (e.g., $\beta_f$ for the smoothness constant, $\mu_f$ for the (PŁ)). Similarly, all constants related to the reduced objective $F$ will be subscripted accordingly (e.g., $\beta_F$, $\mu_F$).*

## 3  Main Results

We now present the main theoretical contributions of this work. Our results formally demonstrate how exploiting known geometric structure at optimality via reduction mappings leads to improved smoothness and sharpness properties, which in turn result in improved convergence rates for gradient-based methods. Among the four equivalent sharpness conditions discussed previously, we will, for the purpose of this section, adopt the (MB) condition as our reference framework for comparing sharpness constants. When discussing convergence rates, we will switch to the (PŁ) condition, which is more directly linked to iteration complexity. For clarity of exposition, we organise the results into two parts: smoothness improvements and improvements of the Morse–Bott constant.

### 3.1  Smoothness Improvement under Affine Reduction Mappings

We first establish that affine reduction mappings, by eliminating alignment with the worst-case curvature directions of the original problem, yield strictly improved smoothness constants for the reduced problem. This result is made precise in the following theorem. A noteworthy special case of affine mappings are constant mappings, whose analysis is deferred to Appendix A.

**Theorem 1 (Sharper Smoothness Constant for Reduced Functions with Affine Mappings)**  *Let $f : \mathbb{R}^n \to \mathbb{R}$ be a $C^2$ function satisfying Assumption 1 on the compact neighbourhood $\mathcal{N}$. Let $\Psi : \mathbb{R}^{n_1} \to \mathbb{R}^{n_2}$ be an affine mapping, and define the reduction mapping $\Phi(x_1) = \big(x_1, \Psi(x_1)\big)$, the reduced function $F$, and the feasible manifold $\mathcal{M}_F$ as in Definitions 2 and 3. We consider the local feasible manifold $\mathcal{M}_F^{\mathrm{loc}} := \mathcal{M}_F \cap \mathcal{N}$.*

*Let $\sigma_1, \ldots, \sigma_n$ be the singular values of $\nabla^2 f(x)$ arranged in descending order. Suppose that, for every $x \in \mathcal{M}_F^{\mathrm{loc}}$, the largest singular value (denoted $\sigma_{\max}$) has multiplicity $p \geq 1$, with associated dominant subspace $\Sigma_{\max}$. Assume the following:*

1. *There exists a uniform constant $\varepsilon \in (0, 1]$ such that for all unit vectors $v \in \Sigma_{\max}$,*
$$\|\mathrm{P}_{\mathrm{T}_x \mathcal{M}_F^{\mathrm{loc}}} v\| \leq 1 - \varepsilon \,,$$
   *where $\mathrm{P}_{\mathrm{T}_x \mathcal{M}_F^{\mathrm{loc}}}$ is the orthogonal projection onto the tangent space $\mathrm{T}_x \mathcal{M}_F^{\mathrm{loc}}$.*

2. *There is a uniform spectral gap:*
$$\Delta_{\max} := \inf_{x \in \mathcal{M}_F^{\mathrm{loc}}} \big[\sigma_{\max}\big(\nabla^2 f(x)\big) - \sigma_{p+1}\big(\nabla^2 f(x)\big)\big] > 0 \,.$$

*Then, equipping $\mathbb{R}^{n_1}$ with the pullback metric induced by the embedding $\Phi$, the Riemannian gradient of $F$ is Lipschitz continuous with constant $\beta_F$ satisfying:*
$$\beta_F \leq \beta_f - \Delta_{\max}(2\varepsilon - \varepsilon^2) < \beta_f \,.$$

**Proof Sketch of Theorem 1**  The proof relies on viewing the smoothness of the reduced function $F$ as the largest curvature of $f$ along the feasible manifold $\mathcal{M}_F^{\mathrm{loc}}$. Since the dominant curvature directions of $f$ are not contained in the tangent space of $\mathcal{M}_F^{\mathrm{loc}}$, the restriction of the Hessian to $\mathcal{M}_F^{\mathrm{loc}}$ exhibits strictly smaller operator norm. By combining this observation with the spectral gap assumption, we obtain a strict improvement in the smoothness constant. The argument is formalised in Appendix C. ∎

This result establishes that the smoothness constant of the reduced function $F$ is strictly smaller than that of the original function $f$, with the improvement governed by the geometric properties of the

intersection between the dominant curvature directions and the feasible manifold. These assumptions correspond to generic properties under mild conditions, as discussed in Appendices E.3 and E.4.

We now translate this result into the standard Euclidean setting, where the role of the pullback metric becomes explicit through the following corollary.

**Corollary 1 (Euclidean Smoothness Bound under Affine Mappings)** *Under the setting of Theorem 1, define the Euclidean smoothness constant of the reduced function $F$ as $\beta_F^{(E)} := \sup_{x_1}\big\|\nabla^2 F\big\|$, where the derivative of $\Phi$ is $\mathrm{D}\,\Phi = \begin{pmatrix} I \\ \mathrm{D}\,\Psi \end{pmatrix}$ and $M^{(\Phi)} := \lambda_{\max}\big(\mathrm{D}\,\Phi^\top \mathrm{D}\,\Phi\big)$. Then,*

$$\beta_F^{(E)} \le M^{(\Phi)} \big[\beta_f - \Delta_{\max}(2\varepsilon - \varepsilon^2)\big] < M^{(\Phi)}\,\beta_f.$$

*In particular, when $\Psi$ is an orthogonal projection, $M^{(\Phi)} = 2$, and the sufficient condition ensuring $\beta_F^{(E)} < \beta_f$ reduces to*

$$\Delta_{\max}(2\varepsilon - \varepsilon^2) > \tfrac{1}{2}\,\beta_f. \tag{$\star$}$$

Condition ($\star$) requires the Hessian of $f$ to exhibit a significant spectral gap between its largest and subsequent eigenvalues. This scenario corresponds to highly anisotropic curvature, where a small number of dominant directions govern the largest curvatures of the loss landscape. Such spectral structures are commonly observed in over-parametrised models in machine learning, where empirical studies have consistently reported Hessians with a few large outlier eigenvalues and a bulk of near-zero eigenvalues [66–70]. This phenomenon supports the practical relevance of our condition in many deep learning settings. We note that the degradation observed in the Euclidean smoothness constant in Corollary 1 arises because the reduction mapping induces a non-Euclidean geometry on the reduced space. The natural metric in this setting is the pullback metric induced by the mapping itself, under which the improvement in smoothness is directly captured, as shown in Theorem 1. We will return to this point and its implications for convergence rates in Section 4.

### 3.2 Smoothness Improvement under Nonlinear Reduction Mappings

We now extend the previous result to the more interesting case where the reduction mapping $\Psi(x_1)$ is nonlinear. In this setting, the feasible manifold $\mathcal{M}_F^{\mathrm{loc}}$ becomes curved within the ambient space, introducing an additional curvature contribution to the reduced function $F$. This contribution is captured by the correction term $\mathrm{C}(x_1)$ in the Hessian (see Lemma 3). Intuitively, this term quantifies the bending effect of $\mathcal{M}_F^{\mathrm{loc}}$ and the extent to which the nonlinearity of $\Psi$ adds curvature to $F$. When this correction remains sufficiently small relative to the spectral gap between the largest curvature directions of $f$ and their restriction to $\mathcal{M}_F^{\mathrm{loc}}$, the reduction in the smoothness constant is preserved. This is formalised in the following theorem and the corresponding corollary for the Euclidean case.

**Theorem 2 (Sharper Smoothness Constant for Reduced Functions with Nonlinear Mappings)** *Let $f$, $\Phi$, $F$, and $\mathcal{M}_F^{\mathrm{loc}}$ be as in Theorem 1, except that $\Psi(x_1)$ is now a general $C^2$ mapping. Assume:*

1. *There exist constants $Q, Z > 0$ such that for all $x_1 \in \mathcal{M}_F^{\mathrm{loc}}$, $\|\mathrm{D}^2\,\Psi(x_1)\| \le Q$ and $\|\nabla_{x_2} f(x_1, \Psi(x_1))\| \le Z$.*

2. *The correction term satisfies $\frac{Q\,Z}{m^{(\Phi)}} < \delta$, where $m^{(\Phi)} = \lambda_{\min}\big(\mathrm{D}\,\Phi^\top \mathrm{D}\,\Phi\big)$ and $\delta$ is the curvature gap defined as*

$$\delta := \sigma_{\max}\big(\nabla^2 f(\Phi(x_1))\big) - \sigma_{\max}\Big(\nabla^2 f(\Phi(x_1))\big|_{\mathrm{T}_{\Phi(x_1)}\mathcal{M}_F^{\mathrm{loc}}}\Big).$$

*Then, the reduced function $F$ has a Lipschitz continuous Riemannian gradient with*

$$\beta_F \le \beta_f - \Delta_{\max}(2\varepsilon - \varepsilon^2) + \frac{Q\,Z}{m^{(\Phi)}} < \beta_f.$$

*In particular, when $\Psi$ is an orthogonal projection, $m^{(\Phi)} = 1$.*

**Proof Sketch of Theorem 2**   The proof extends the affine case by accounting for the additional curvature induced by the nonlinearity of $\Psi$. This is captured by the correction term $C(x_1)$ in the Hessian of $F$, which arises due to the curvature of the feasible manifold $\mathcal{M}_F^{\mathrm{loc}}$. By controlling $\|C(x_1)\|$ via bounds on $D^2 \Psi$ and $\nabla_{x_2} f$, and ensuring that it remains strictly smaller than the curvature gap $\delta$ obtained from the projection step, we show that the overall smoothness constant of $F$ is still strictly lower than that of $f$. The result follows by combining these bounds and applying Weyl's inequality for perturbed operators. Full details are given in Appendix C.                    ∎

**Corollary 2 (Euclidean Smoothness Bound under Nonlinear Mappings)**   *Under the setting of Theorem 2, the Euclidean smoothness constant of the reduced function $F$ satisfies*

$$\beta_F^{(E)} \leq M^{(\Phi)} \left[ \beta_f - \Delta_{\max}(2\varepsilon - \varepsilon^2) \right] + Q\, Z < M^{(\Phi)}\, \beta_f \,,$$

*where $M^{(\Phi)}$ is the metric distortion factor defined as in Corollary 1. In particular, when $\Psi$ is an orthogonal projection, $M^{(\Phi)} = 2$, and the sufficient condition ensuring $\beta_F^{(E)} < \beta_f$ simplifies to*

$$\Delta_{\max}(2\varepsilon - \varepsilon^2) > \tfrac{1}{2}\left( \beta_f + Q\, Z \right).$$

Reduction mappings often naturally arise in bilevel optimisation settings, where $x_2 = \Psi(x_1)$ is implicitly defined as the solution to an inner problem. Depending on the problem structure, these mappings can be affine or nonlinear, and our results apply equally in both cases. We formalise this in the remark below.

**Remark 1 (Inner Mappings as Argmin Problems)**   *Reduction mappings often arise when $\Psi(x_1)$ is defined implicitly as a local solution to an inner optimisation problem*

$$\Psi(x_1) \in \arg\min_{u \in \mathcal{C}} G(x_1, u) \,.$$

*Under standard regularity conditions, such as constraint qualifications and strict second-order sufficiency (SSOSC), classical sensitivity results ensure that $\Psi(x_1)$ is locally $C^2$ [71]. Thus, our previous theorems for affine and nonlinear mappings apply directly in such settings.*

To better understand the bounds on the correction term, when the mapping $\Psi$ arises from an inner `argmin` problem, the constants $Q$ and $Z$ have been explicitly quantified in Theorem 4 in Appendix D.

### 3.3   Morse–Bott Constant Improvement under Reduction Mappings

In Appendix D, we establish that if the original function $f$ satisfies the (MB) property, then the reduced function $F$ obtained via reduction mappings also satisfies the (MB) property, albeit potentially with a different constant. In the following theorem, we strengthen this result by showing that the (MB) constant of the reduced problem is in fact strictly improved compared to that of the original problem.

**Theorem 3 (Strict Improvement of the Morse–Bott Constant under Smooth Reduction)**   *Let $f : \mathbb{R}^n \to \mathbb{R}$ be a $C^2$ function where the solution manifold $\mathcal{S}$ satisfies the $\mu_f$-MB property within a compact neighbourhood $\mathcal{N}$. Let $\Psi : \mathbb{R}^{n_1} \to \mathbb{R}^{n_2}$ be a $C^2$ mapping and define the reduced function $F(x_1) = f(x_1, \Psi(x_1))$, with local feasible manifold $\mathcal{M}_F^{\mathrm{loc}} = \mathcal{M}_F \cap \mathcal{N}$.*

*At each $\bar{x} \in \mathcal{S} \cap \mathcal{M}_F^{\mathrm{loc}}$, let $H_{\bar{x}}$ denote the restriction of $\nabla^2 f(\bar{x})$ to $N_{\bar{x}}\mathcal{S}$. Assume:*

1. *The smallest eigenvalue $\lambda_{\min}(H_{\bar{x}})$ has multiplicity $m$, with eigenspace $E_{\min}$, and for all $v \in E_{\min}$, $\|v\| = 1$,*
$$\|P_{T_{\bar{x}}\mathcal{M}_F^{\mathrm{loc}}}\, v\| \leq 1 - \varepsilon \,,$$
   *holds uniformly for some $\varepsilon \in (0, 1]$.*
2. *The spectral gap $\Delta_{\min} := \inf_{\bar{x} \in \mathcal{S} \cap \mathcal{M}_F^{\mathrm{loc}}} \left[ \lambda_{n-m}(H_{\bar{x}}) - \lambda_{\min}(H_{\bar{x}}) \right] > 0.$*

*Then, for the (MB) property of the reduced function $F$, we obtain a strictly improved constant*

$$\mu_F \geq \mu_f + \Delta_{\min}(2\varepsilon - \varepsilon^2) > \mu_f \,.$$

**Proof** The result follows by applying the same geometric argument as in Theorem 1, now to the positive definite Hessian $H_{\bar{x}}$ restricted to $N_{\bar{x}}\mathcal{S}$. Since we operate entirely within the normal space and on the solution manifold, only eigenvalues and eigenspaces matter, and the nonlinearity of $\Psi$ has no effect. ∎

In the Euclidean setting, the improvement in the Morse–Bott constant is further scaled by the pullback metric. Specifically, since the Euclidean constant is related to the intrinsic pullback constant via $\mu_F^{(E)} \geq m^{(\Phi)}\mu_F$, the distortion introduced by the metric, through its smallest eigenvalue $m^{(\Phi)} \geq 1$, amplifies the effective (MB) constant. This shows that the pullback metric does not only improve the intrinsic condition number but also results in a better constant when measured under the Euclidean metric. In particular, when $\Psi$ is a projection mapping, we have $m^{(\Phi)} = 1$, and the Euclidean result coincides exactly with the pullback metric case.

**Remark 2 (Morse–Bott Constant Equivalence [64])** *Since in Theorem 3 we showed that $\mu_F > \mu_f$ for the (MB) property, by equivalence it follows that the constants $\mu$ for the (PŁ), (EB), and (QG) conditions will follow the same strict inequality.*

## 4  Convergence Rate Results and Discussion

In this section, we discuss the algorithmic implications of our geometric analysis of reduction mappings. In particular, we focus on the impact of the improved smoothness and (MB)—and by equivalence (PŁ)—constants obtained in Theorems 1, 2, and 3 on the convergence behaviour of first-order methods.

When the reduced problem is equipped with the pullback metric induced by the reduction mapping $\Phi$, our results show that the condition number of the reduced function $F$ is strictly improved relative to that of the original problem $f$. This theoretical gain directly translates into faster local linear convergence rates when applying geometrically preconditioned gradient descent (GeoPrecGD) on the reduced objective. We formally state this result below. A broader discussion on metric choices, algorithmic implications, and practical considerations follows.

**Corollary 3 (Faster Linear Convergence of the Reduced Function)** *Under the settings of Theorems 1, 2, and 3, equipping $\mathbb{R}^{n_1}$ with the pullback metric induced by $\Phi$ yields a strictly improved condition number for the reduced problem*

$$\kappa_F := \frac{\beta_F}{\mu_F} < \frac{\beta_f}{\mu_f} =: \kappa_f\,.$$

*As a result, preconditioned gradient descent applied to $F$ achieves a strictly faster local linear convergence rate under (PŁ) condition compared to gradient descent on $f$ [5]. Specifically:*

- *The rate factor improves from $\mathcal{O}(\exp(-t/\kappa_f))$ to $\mathcal{O}(\exp(-t/\kappa_F))$.*
- *The iteration complexity to achieve accuracy $\epsilon$ improves from $\mathcal{O}(\kappa_f \log(1/\epsilon))$ to $\mathcal{O}(\kappa_F \log(1/\epsilon))$.*

Below, we make explicit the form of the geometrically preconditioned gradient descent algorithm applied to the reduced problem $F$ under the pullback metric induced by $\Phi$

$$x_1^{(t+1)} = x_1^{(t)} - \eta\,R^{-1}\,\nabla F\left(x_1^{(t)}\right), \quad \text{with} \quad R := D\,\Phi^\top D\,\Phi\,. \qquad \text{(GeoPrecGD)}$$

This method performs steepest descent under the pullback metric, aligning the descent directions with the intrinsic geometry induced by the reduction mapping—a fundamental principle in optimisation [72]. While preconditioning introduces additional computational cost due to the inversion of $R$, such overhead can often be mitigated by exploiting the structure of $R$, as shown in prior works [73, 74, 50, 75, 76, 52, 51]. In particular, for affine mappings, $R$ is constant and cheap to apply, whereas for nonlinear mappings, structured or approximate solvers can be used. Designing efficient implementations of these methods is primarily mapping dependent and beyond the scope of this work, where our focus is on the iteration complexity benefits arising from the improved conditioning.

An instructive special case is when $\Psi$ is an orthogonal projection; a common choice for many reparametrisations. If $\Psi$ is affine, then $R$ simplifies, and its inverse is given by

$R^{-1} = I - \frac{1}{2} \, \mathrm{D} \, \Psi^\top \, \mathrm{D} \, \Psi$, implying no additional computational overhead. More generally, if $\Psi$ is nonlinear but still defines an orthogonal projection onto a manifold, the situation remains favourable, particularly near convergence in the defined neighbourhood. As iterates approach the solution manifold, the derivative $\mathrm{D} \, \Phi$ locally behaves as a linear orthogonal projection onto the tangent space of the solution manifold, with the condition number of $R$ remaining close to two. This favourable and stable spectrum near the solution means the preconditioned linear system within the (GeoPrecGD) step ($Rz = \nabla F$) can be solved efficiently. Specifically, iterative solvers (*e.g.*, Conjugate Gradient) applied to this system benefit from rapid convergence due to the low condition number. Also, the structure of $R$ allows for an efficient and cheap inversion via the Woodbury formula, depending on whether $n_1 > n_2$ or vice versa.

We validate our theory with synthetic experiments in Appendix G. Notably, for quadratic objectives, our geometrically preconditioned gradient descent is equivalent to the full Newton method on the reduced problem; and for nonlinear least-squares, it is equivalent to the Gauss–Newton method applied to the reduced problem. Thus, our approach recovers Newton and Gauss–Newton as special cases while extending beyond them to more general settings.

While preconditioning is necessary and recommended [53] to fully exploit the improved geometry induced by the reduction mapping, improvements can still be observed under Euclidean gradient descent. Specifically, if the condition ($\star$) holds, we still have $\beta_F^{(E)} < \beta_f$, leading to a better condition number even without preconditioning. Even if this condition fails, the improvement in the sharpness constant $\mu_F^{(E)} > \mu_f$ may still result in a better overall condition number $\kappa_F^{(E)}$ compared to $\kappa_f$. Thus, gains are still possible, although they may be limited.

If a practitioner desires additional control over the condition number in Euclidean space for convergence guarantees, the reduction mapping can be modified to induce approximate isometries via a simple *small-slope method*. This approach is general and may warrant a more systematic study; however, a full investigation of such strategies lies outside the scope of this paper. We sketch the idea below.

**Remark 3 (Small-Slope Method and Approximate Isometry)** *When using reduction mappings, the pullback metric takes the form of $R = I + \mathrm{D} \, \Psi^\top \, \mathrm{D} \, \Psi$, which in general is not identity, leading to potential degradation in the Euclidean smoothness constant proportional to $\lambda_{\max}(R)$. To mitigate this, we propose a simple* small-slope method, *where the mapping is rescaled as $\Psi_\alpha(x_1) := \alpha \Psi(x_1)$ for a small $\alpha > 0$. This modifies the metric to $R_\alpha = I + \alpha^2 \, \mathrm{D} \, \Psi^\top \, \mathrm{D} \, \Psi$, whose condition number becomes*

$$\kappa(R_\alpha) = \frac{1 + \alpha^2 \lambda_{\max}(\mathrm{D} \, \Psi^\top \, \mathrm{D} \, \Psi)}{1 + \alpha^2 \lambda_{\min}(\mathrm{D} \, \Psi^\top \, \mathrm{D} \, \Psi)} .$$

*Thus by making $\alpha$ sufficiently small, $R_\alpha$ becomes approximately isometric, improving the likelihood that $\beta_F^{(E)} \lesssim \beta_f$. This method offers a simple yet effective trade-off between improving conditioning and preserving geometric fidelity. Notably, if the geometric structure encoded by $\Psi$ is not scale-invariant, excessively small $\alpha$ may distort the mapping, potentially degrading its ability to capture the correct geometry of the $x_2$ variables. This trade-off is less critical in many common cases—such as subspaces, cones, or orthogonal structures—where the geometry is inherently scale-invariant. The parameter $\alpha$ can be treated as a user-defined knob, balancing conditioning improvement against the precision of the reduction.*

## 4.1 Limitations and Future Directions

Our analysis assumes that the reduction mapping $\Psi$ can be evaluated exactly. When $\Psi$ is defined implicitly, this idealisation neglects the approximation errors that may arise in practice, potentially introducing bias in the reduced gradients and weakening the theoretical convergence guarantees. Moreover, the computational cost of evaluating $\Psi(x_1)$ is inherently problem-specific and is not explicitly captured in the iteration complexity analysis. A rigorous treatment that jointly considers the benefits of reduction mappings and their evaluation cost remains an open, problem-dependent direction for future work.

Beyond these considerations, our analysis is confined to local properties and deterministic first-order methods. Extending the theory to global convergence settings, particularly in nonconvex landscapes, is an important future direction. In this context, it would be valuable to explore how

reduction mappings reshape the landscape globally, including their potential to eliminate or introduce saddle points and spurious minima, extending the framework of [41]. Another promising direction is applying our geometric framework to specific structured problems where the objective admits compositional forms, such as matrix factorisation, nonlinear regression, or neural network training, to obtain practical insights. Finally, extending the analysis to stochastic settings, for example by studying stochastic gradient descent (SGD) under reduction mappings and characterising its regularity conditions, would enhance the relevance of our results in large-scale applications.

## 5 Conclusion

In this work, we presented a unified geometric framework for analysing reduction mappings in optimisation problems with structured solution sets. By explicitly incorporating mappings that encode known geometric structures at optimality, we showed that the resulting reduced problems exhibit strictly improved smoothness and sharpness properties. This leads to enhanced local condition numbers and provably faster convergence rates when applying appropriately preconditioned first-order methods. Our analysis generalises seamlessly from affine to nonlinear mappings, carefully accounting for the additional curvature induced by the bending of the feasible manifold. While the improvements are fundamentally intrinsic to the pullback geometry, we further showed that, under appropriate constructions and trade-offs, these gains can also manifest in the Euclidean metric. Throughout, we emphasised the generality and flexibility of our framework, illustrating that reduction mappings may be given explicitly or arise implicitly as solutions to inner optimisation problems, thereby connecting our approach to classical bilevel and composition formulations via implicit differentiation. We believe this geometric perspective offers a principled and broadly applicable toolset for designing more efficient optimisation algorithms that better exploit problem structure, with potential impact across areas such as matrix factorisation, deep learning, and other structured nonconvex problems.

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

# A  A Gentle Start

To build intuition for the broader theoretical developments in this work, we begin with two illustrative examples. Each example is carefully chosen to highlight a different geometric structure of the minimisers: a function with a unique isolated minimum and a function with a continuum of non-isolated minima. For each case, we examine the curvature of the landscape through the lens of the Hessian and its maximal eigenvalue, offering insight into the local geometry.

We then apply a range of reduction mappings to these examples—each designed to restrict the optimisation to a lower-dimensional subspace or manifold—and observe how these mappings affect the curvature of the objective function. Of particular interest is how the maximum curvature behaves under these reductions, as this has direct implications for the conditioning of the problem and the performance of iterative methods.

The section concludes with a focused analysis of constant mappings, a special class where the reduction ignores part of the variable space entirely. This case serves as a useful analytical baseline, offering contrast to structured mappings that actively exploit the problem geometry, whereas constant mappings remain agnostic to how this structure influences the reduced variables.

## A.1  Example 1: Single Isolated Minimum

Consider the function $f : \mathbb{R}^2 \to \mathbb{R}$ defined by

$$f(x_1, x_2) = x_1^2 + M(x_2 - x_1)^2 \,, \tag{6}$$

where $M \gg 1$ is a large constant. This function exhibits strong anisotropy: the quadratic term in $x_2$ introduces high curvature along the $x_2$-direction. As a result, the Lipschitz constant of the gradient (*i.e.*, the smoothness constant) $\beta_f$ is on the order of $M$, reflecting the steep variations in the landscape. We note that this function is used in Figure 1 with $M = 2$ for ease of visualisation.

To make this explicit, we compute the gradient and Hessian of $f$ as follows

$$\nabla f(x_1, x_2) = \begin{pmatrix} 2x_1 + 2M(x_1 - x_2) \\ 2M(x_2 - x_1) \end{pmatrix}, \qquad \nabla^2 f(x_1, x_2) = \begin{pmatrix} 2 + 2M & -2M \\ -2M & 2M \end{pmatrix} . \tag{7}$$

The eigenvalues of the Hessian are given by

$$\lambda = \frac{(2 + 4M) \pm \sqrt{(2 + 4M)^2 - 16M}}{2} \,. \tag{8}$$

If one instead fixes $x_2$ via a constant mapping that lies on the solution manifold, the effective maximum curvature in the $x_1$-direction becomes $2 + 2M$, since $x_2$ is no longer allowed to adapt to changes in $x_1$, and the mismatch induces high curvature.

Now consider an affine mapping, which can arise from a bilevel setting such that

$$x_2^*(x_1) = \arg\min_{x_2} f(x_1, x_2) \,. \tag{9}$$

For the function above, it is easy to verify that the optimal inner solution satisfies $x_2^*(x_1) = x_1$. Along the reduced trajectory $(x_1, x_2^*(x_1)) = (x_1, x_1)$, the function simplifies to

$$F(x_1) = f(x_1, x_2^*(x_1)) = x_1^2 \,. \tag{10}$$

In this formulation, the steep curvature contributed by the term $M(x_2 - x_1)^2$ is eliminated entirely, and the resulting function $F$ has a smoothness constant that is independent of $M$.

This example illustrates that allowing $x_2$ to adjust optimally can "iron out" steep variations that would otherwise hinder optimisation. However, this is true only when the mapping is well-designed. To see that not all reductions lead to improved curvature, consider instead a nonlinear mapping given by

$$x_2(x_1) = x_1 - 2\sin(x_1) \,. \tag{11}$$

Under this mapping, the reduced function becomes

$$F(x_1) = x_1^2 + 4M \sin^2(x_1) \,. \tag{12}$$

The curvature of this reduced function is now influenced by the oscillatory term $\sin^2(x_1)$, and its second derivative includes a component proportional to $4M\cos(2x_1)$, leading to increased curvature compared to the original function $f$.

This contrasting case makes clear that not all mappings are beneficial: poorly chosen or misaligned mappings can introduce new sources of curvature rather than mitigate them. Hence, curvature reduction is not an automatic consequence of reparametrisation—it depends critically on the mapping being well-aligned with the geometry of the solution manifold and its effect on the optimisation trajectory. In well-structured settings, such as when $x_2^*(x_1)$ is derived from minimising over $x_2$, the resulting outer function $F$ can exhibit substantially improved curvature characteristics.

In Figure 2, we plot the curvature

$$\kappa(x) = \frac{f''(x)}{\left[1 + (f'(x))^2\right]^{3/2}}, \tag{13}$$

over the domain $x \in [-0.5, 0.5]$ with $M = 10$ for the reduced cases. For comparison, we also indicate the corresponding maximum curvature value (*i.e.*, the largest eigenvalue of the Hessian) for each case, including the original 2D function.

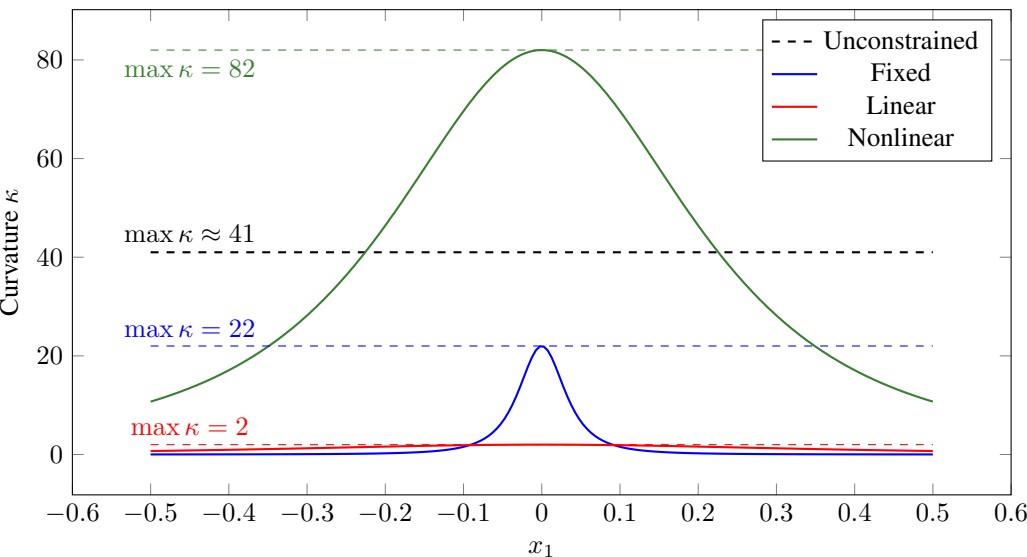

Figure 2: Curvature profiles for three reduction mappings. The full (unconstrained) function $f(x_1, x_2) = x_1^2 + 10(x_2 - x_1)^2$ exhibits high curvature (approximately 41). Fixing $x_2 = 0$ yields $F_{\text{fixed}}(x_1) = 11x_1^2$, with curvature 22 at $x_1 = 0$. The bilevel reduction $F_{\text{linear}}(x_1) = x_1^2$ has lower curvature (2 at $x_1 = 0$), while the nonlinear mapping induces strong oscillations and increases the maximum curvature to 82.

## A.2 Example 2: Non-Isolated Minima

Consider the function $f : \mathbb{R}^2 \to \mathbb{R}$ defined by

$$f(x_1, x_2) = \varphi(x_1) + (x_2 - \sin(x_1))^2, \tag{14}$$

where the flat-bottom quartic $\varphi$ is defined as

$$\varphi(x_1) := \begin{cases} (x_1 - \alpha)^4, & x_1 < \alpha, \\ 0, & x_1 \in [\alpha, \beta], \\ (x_1 - \beta)^4, & x_1 > \beta. \end{cases} \tag{15}$$

The gradient of $f$ is given by

$$\nabla f(x_1, x_2) = \begin{pmatrix} \varphi'(x_1) - 2(x_2 - \sin(x_1))\cos(x_1) \\ 2(x_2 - \sin(x_1)) \end{pmatrix}, \tag{16}$$

where

$$\varphi'(x_1) = \begin{cases} 4(x_1 - \alpha)^3, & x_1 < \alpha \,, \\ 0, & x_1 \in [\alpha, \beta] \,, \\ 4(x_1 - \beta)^3, & x_1 > \beta \,. \end{cases} \tag{17}$$

The Hessian is

$$\nabla^2 f(x_1, x_2) = \begin{bmatrix} \varphi''(x_1) + 2(x_2 - \sin(x_1))\sin(x_1) + 2\cos^2 x_1 & -2\cos(x_1) \\ -2\cos(x_1) & 2 \end{bmatrix}, \tag{18}$$

where

$$\varphi''(x_1) = \begin{cases} 12(x_1 - \alpha)^2, & x_1 < \alpha \,, \\ 0, & x_1 \in [\alpha, \beta] \,, \\ 12(x_1 - \beta)^2, & x_1 > \beta \,. \end{cases} \tag{19}$$

Since the Hessian is symmetric, its eigenvalues are real and given by

$$\lambda_\pm = \frac{1}{2}\left(a + c \pm \sqrt{(a-c)^2 + 4b^2}\right), \tag{20}$$

where $a = \nabla^2_{x_1 x_1} f$, $b = \nabla^2_{x_1 x_2} f = -2\cos(x_1)$, and $c = \nabla^2_{x_2 x_2} f = 2$. These expressions follow from the standard characteristic equation for symmetric $2 \times 2$ matrices. In our analysis, we record the largest eigenvalue of the Hessian restricted to the region $x_1 \in [\alpha, \beta]$, where $\varphi''(x_1) = 0$.

We consider three reduction mappings $\Psi_i : \mathbb{R} \to \mathbb{R}$, each inducing a reduced function $F_i(x_1) = f(x_1, \Psi_i(x_1))$. The quality of each mapping can be assessed by the curvature of the resulting reduced function, particularly over the flat region $[\alpha, \beta]$, where $\varphi''(x_1) = 0$.

- **Nonlinear (Implicit) mapping:** $\Psi_1(x_1) = \sin(x_1)$. This choice corresponds to the exact solution of the inner minimisation problem

$$\Psi_1(x_1) = \arg\min_{x_2} f(x_1, x_2), \tag{21}$$

  and cancels the coupling term, *i.e.*, $(x_2 - \sin(x_1))^2 = 0$. The resulting reduced function is

$$F_1(x_1) = \varphi(x_1), \tag{22}$$

  which is piecewise quartic with a flat basin over $[\alpha, \beta]$. Since both the first and second derivatives vanish in that interval, $F_1$ has zero curvature there. This mapping represents the ideal scenario for dimensionality reduction, as it perfectly aligns with the optimal structure of the original function.

- **Fixed mapping:** $\Psi_2(x_1) = 0$. This mapping treats $x_2$ as a constant and neglects the structure of the inner problem. The reduced function becomes

$$F_2(x_1) = \varphi(x_1) + \sin^2(x_1), \tag{23}$$

  where the additional term $\sin^2(x_1)$ introduces artificial curvature even inside the flat region, due to the mismatch between the fixed choice $x_2 = 0$ and the optimal one $x_2 = \sin(x_1)$. While simple to implement, this mapping results in a highly curved landscape that can significantly hinder optimisation.

- **Linear mapping:** $\Psi_3(x_1) = x_1$. This mapping is a linear approximation to the sine function near the origin and yields

$$F_3(x_1) = \varphi(x_1) + (x_1 - \sin(x_1))^2. \tag{24}$$

  Inside the interval $[\alpha, \beta]$, $\varphi'' = 0$, but the additional term $(x_1 - \sin(x_1))^2$ introduces nonzero curvature. However, unlike the fixed mapping, the misalignment with $\sin(x_1)$ is smaller, especially near the origin. Thus, $F_3$ achieves lower curvature than $F_2$, but does not eliminate coupling completely. It represents a computationally cheap, yet structurally-informed approximation.

These three reductions are visualised in Figure 3, where each curve lies on the surface of $f$ and projects to its corresponding manifold $\mathcal{M}_{F_i}$ in the base plane. Figure 4 plots the curvature $\kappa(x_1)$ of each reduction and confirms that the implicit mapping achieves the smallest curvature, which vanishes on $[\alpha, \beta]$, where we have picked $\alpha = -0.5$ and $\beta = 0.5$.

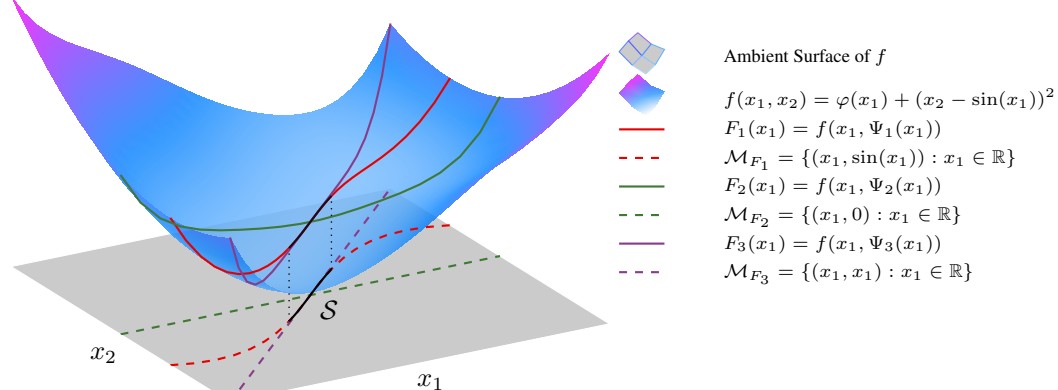

Figure 3: Visualisation of the ambient function $f(x_1, x_2) = \varphi(x_1) + (x_2 - \sin(x_1))^2$, with $\varphi$ a flat-bottom quartic, and three lifted restriction curves corresponding to different reduction mappings $\Psi_i(x_1)$. The black curve $\mathcal{S}$ denotes the global minimisers of $f$, forming a non-isolated solution manifold parametrised by $x_1 \in [-0.5, 0.5]$ and $x_2 = \sin(x_1)$. Each lifted curve lies on the surface $f$, and the dashed curves on the base plane show the image of each corresponding manifold $\mathcal{M}_{F_i}$.

### A.3 Constant Mappings: A Special Case

As a special case of our reduction framework, consider a constant mapping of the form

$$\Psi(x_1) = \bar{x}_2, \quad \text{for some fixed } \bar{x}_2 \in \mathbb{R}^{n_2}. \tag{25}$$

The reduced function is then

$$F(x_1) = f(x_1, \bar{x}_2), \tag{26}$$

with reparametrisation map $\Phi(x_1) = (x_1, \bar{x}_2)$, whose Jacobian satisfies

$$\mathrm{D}\,\Phi(x_1) = \begin{pmatrix} I_{n_1} \\ 0 \end{pmatrix} \in \mathbb{R}^{(n_1+n_2)\times n_1}, \quad \mathrm{D}\,\Phi(x_1)^\top \mathrm{D}\,\Phi(x_1) = I_{n_1}. \tag{27}$$

This isometric property ensures that the geometry of the reduced space is Euclidean, and hence optimisation proceeds without distortion or the need for metric preconditioning.

Although the mapping is static, it completely eliminates the $x_2$-directions. If the original function $f$ exhibits strong curvature in those directions, the reduced function $F$ may be significantly better conditioned. In this way, constant mappings can yield nontrivial curvature reduction, even without adapting to the structure of $f$ as $x_1$ varies.

Under standard assumptions (*e.g.*, smoothness and PŁ conditions), the reduced function $F$ can inherit the same convergence guarantees as affine mappings, provided the non-tangency condition and a uniform spectral gap hold. The key advantage here is that these guarantees apply directly in the Euclidean setting, without requiring any geometric correction.

The main limitation of constant mappings lies in their lack of adaptivity. Since the mapping is fixed, it cannot adjust to the optimal choice of $x_2^*(x_1)$ as $x_1$ evolves during optimisation. In settings where $x_1$ and $x_2$ are tightly coupled, this can lead to misalignment between the reduced function $F(x_1) = f(x_1, \bar{x}_2)$ and the true solution manifold of $f$. While $\bar{x}_2$ may belong to the global minimiser set of $f$, it need not remain optimal for all values of $x_1$. As a result, $F$ may fail to faithfully reflect the structure of $f$ away from neighbourhoods where $(x_1, \bar{x}_2) \in \mathcal{S}$, potentially slowing convergence or yielding suboptimal solutions.

This scenario frequently arises in overparametrised models with symmetric minimiser sets. For instance, in deep neural networks, the optimal classifier weights often lie in the set of Equiangular Tight Frames (ETFs). A constant mapping implicitly selects a particular representative $\bar{x}_2$ in this set and holds it fixed throughout training. While this preserves global optimality in principle, it breaks

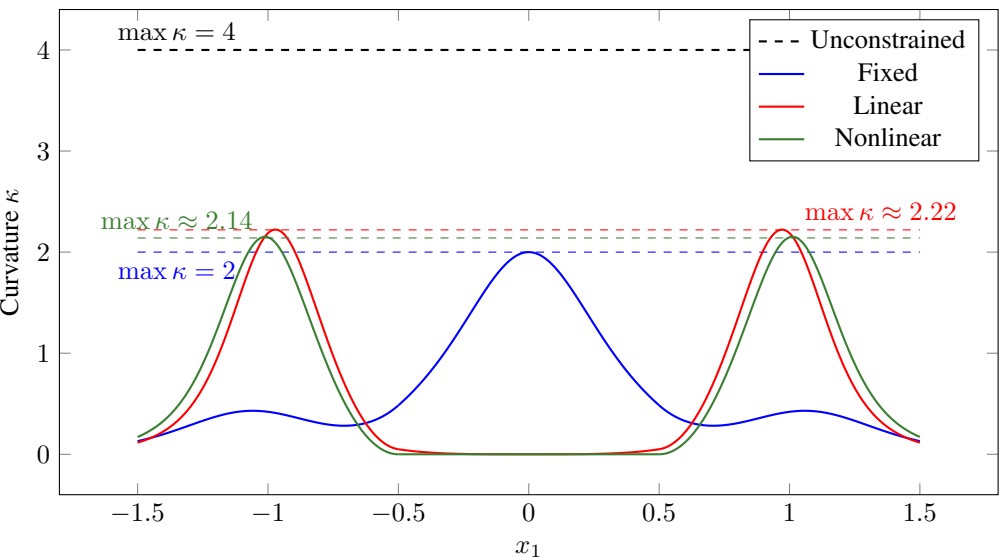

Figure 4: Curvature comparison across different reduction mappings applied to the function $f(x_1, x_2) = \varphi(x_1) + (x_2 - \sin(x_1))^2$, where $\varphi$ is flat on $[-0.5, 0.5]$ and quartic outside. The black dashed line represents the maximum eigenvalue of the full Hessian when no reduction is applied. The blue curve shows the curvature induced by fixing $x_2 = 0$, resulting in high curvature due to mismatch with the optimal $x_2 = \sin(x_1)$. The red curve corresponds to the linear reduction $x_2 = x_1$, which aligns more closely with $\sin(x_1)$ and yields lower curvature. The green curve represents the bilevel (implicit) reduction $x_2 = \sin(x_1)$, which eliminates coupling and flattens the function inside the flat region, driving curvature to zero. Each dashed horizontal line indicates the maximum curvature for its corresponding method.

the symmetry in a way that may be incompatible with the evolution of $x_1$. The reduced function $F$ thus becomes a projection of $f$ onto a fixed slice of the solution manifold, which may not be well-aligned with the optimisation trajectory unless additional structure or coordination is imposed.

In summary, constant mappings offer a geometrically clean and computationally simple reduction strategy. They can yield strong local convergence when well-aligned with the solution structure, and are especially attractive in models where redundant variables can be safely eliminated. However, their rigidity makes them less suitable for problems requiring global coordination between variables or adaptivity along the optimisation path. In practice, they serve as a valuable analytic baseline and an effective modelling choice when the structure of the solution is known or can be fixed a priori.

## B   Extended Related Work

Reparametrisations are widely adopted to exploit problem structure and improve the conditioning of nonconvex optimisation problems [41]. Our framework offers a unified lens to view these diverse approaches as special cases of reduction mappings—whether constant, affine, nonlinear, explicitly defined, or arising implicitly—and systematically analyses their effect on problem conditioning and the convergence of first-order methods. Below, we outline several representative cases.

**Normalisation and induced feature geometry.**   A canonical example is normalisation layers, such as batch normalisation [42] or layer normalisation [43], which perform explicit nonlinear mappings that constrain the scale and centring of activations or parameters. These reparametrisations have been shown to accelerate convergence by implicitly altering the smoothness and sharpness constants [44]. In our framework, this can be interpreted as introducing a reduction mapping $\Psi$ that enforces a fixed feature geometry on specific variables, such as unit variance or zero mean.

**Neural collapse and ETF reparametrisations.**   In the context of neural collapse, our framework naturally captures both prevalent approaches. Fixing the classifier to an equiangular tight frame (ETF)

corresponds to a constant reduction mapping that constrains the classifier parameters to a predefined geometric structure [45]. Alternatively, dynamically solving for the nearest ETF given the current features has been shown to accelerate convergence [46], and fits within our framework as a nonlinear or implicitly defined mapping, enforcing optimal alignment with the feature space. Such reduction mappings can also been extended to regression settings [77]. In all cases, our analysis applies directly, providing theoretical guarantees on improved conditioning and convergence.

**Optimisation on quotient manifolds and gauge fixing.**  In optimisation over quotient manifolds [47, 48], the goal is to account for known invariances by operating directly on equivalence classes, such as optimising over the Grassmannian or Stiefel quotient manifolds. While these approaches formally respect the problem's intrinsic symmetries, they can incur significant per-iteration costs due to expensive manifold operations like projections and retractions. Our framework provides an alternative by enabling explicit gauge fixing via reduction mappings, where one selects a representative from each equivalence class—for example, fixing scaling or orthogonal symmetry through a carefully designed mapping $\Psi$. This strategy can be viewed as a symmetry-breaking mechanism that reduces the optimisation problem to a lower-dimensional space equipped with the pullback metric induced by the reduction mapping. This allows standard first-order methods to operate more efficiently while still respecting the problem's local geometry. Thus, our approach can potentially offer a lightweight and practical alternative to quotient manifold optimisation, particularly suitable for large-scale or deep learning scenarios where iteration efficiency is paramount.

**Preconditioned methods and metric-aware optimisation.**  Preconditioning is a classical approach in optimisation and numerical linear algebra to accelerate convergence by modifying the problem geometry through a change of metric [78]. This perspective underpins variable metric and quasi-Newton methods [78], where gradient steps are taken relative to a preconditioner that improves conditioning. In deep learning, natural gradient descent [49] formalises this idea by using the Fisher information matrix as a Riemannian metric aligned with the model's statistical structure, with recent works showing that such methods can achieve accelerated convergence in overparametrised neural networks [50]. Similarly, recent efforts have revisited scalable preconditioning strategies, such as Kronecker-factored methods [79], Shampoo [51], and Muon [52], which exploit problem structure to efficiently approximate second-order information in large-scale settings. More broadly, the geometry of parameter spaces under reparametrisation has been studied to understand how such choices influence conditioning and learning dynamics [53]. Our framework complements and generalises these approaches by introducing reduction mappings as a geometric preconditioning mechanism, where the pullback metric induced by $\Phi$ serves as an intrinsic, problem-adaptive preconditioner. This allows first-order methods to exploit problem geometry efficiently, without explicit second-order computations, while benefiting from strictly improved condition numbers. Moreover, our analysis applies uniformly to both affine and nonlinear mappings, unifying classical preconditioning with modern geometry-aware methods and extending their reach to a broader class of structured nonconvex problems.

**Reduction mappings and problem dimensionality reduction.**  Classical strategies for problem simplification often rely on variable elimination techniques, such as projection methods, partial optimisation, or block coordinate descent, where certain variables are explicitly or implicitly optimised out to reduce problem dimensionality [54]. A more general formulation arises in bilevel optimisation, where the inner problem implicitly defines a mapping from the outer variables to the lower-level solution [55]. In these settings, implicit differentiation offers a principled mechanism to compute gradients of the reduced objective, effectively collapsing the problem to the outer variables alone. Our framework generalises and unifies these approaches by viewing them as reduction mappings—whether explicitly defined or arising from inner problems solved to optimality—and providing a geometric lens to analyse their impact on problem conditioning. Crucially, our focus extends beyond mere dimensionality reduction, offering sharp characterisations of how these mappings improve local smoothness and sharpness constants of the reduced problem, and thus enhance the convergence rates of first-order methods.

# C   Proof of Main Theorems

This section contains the complete proofs of the main results stated in the paper. Each proof is presented in a separate subsection for clarity. Some of the arguments rely on technical lemmas, which are deferred to Appendix E for readability.

## C.1   Proof of Theorem 1

**Proof**   Since $F(x_1) = f(x_1, \Psi(x_1))$, we have the *reduction mapping*

$$\Phi(x_1) := \begin{pmatrix} x_1 \\ \Psi(x_1) \end{pmatrix} \in \mathbb{R}^{n_1 + n_2}\,.$$

By the chain rule, the gradient of $F$ is given by

$$\nabla F(x_1) = \mathrm{D}\,\Phi(x_1)^\top \nabla f\big(\Phi(x_1)\big) \in \mathbb{R}^{n_1}\,,$$

where the Jacobian of $\Phi$ has the form

$$\mathrm{D}\,\Phi(x_1) = \begin{pmatrix} I_{n_1} \\ J(x_1) \end{pmatrix} \in \mathbb{R}^{(n_1+n_2)\times n_1}, \quad \text{with } J(x_1) := \mathrm{D}\,\Psi(x_1) \in \mathbb{R}^{n_2 \times n_1}\,,$$

and the gradient of $f$ is $\nabla f\big(\Phi(x_1)\big) \in \mathbb{R}^{n_1 + n_2}$.

We define a Riemannian metric on $\mathbb{R}^{n_1}$ induced by $\Phi$ (*i.e.*, a pullback metric) as

$$g(u, v) := \langle u, v \rangle_R = u^\top R\,v, \quad \text{with} \quad R := \mathrm{D}\,\Phi(x_1)^\top \mathrm{D}\,\Phi(x_1)\,.$$

Because the solution mapping $\Psi(x_1)$ is assumed affine, $R$ is constant. Also, we have $\mathrm{D}^2\,\Psi(x_1) = 0$, so no additional correction term appears in the Hessian of $F$.

Differentiating $\nabla F(x_1)$ with respect to $x_1$ yields (see Lemma 3)

$$\nabla^2 F(x_1) = \mathrm{D}\,\Phi(x_1)^\top \nabla^2 f\big(\Phi(x_1)\big)\,\mathrm{D}\,\Phi(x_1)\,.$$

Define $A := \nabla^2 f\big(\Phi(x_1)\big)$ and note that $A|_{\mathcal{V}} := \nabla^2 F(x_1)$ is the restriction of $A$ to the $n_1$-dimensional subspace

$$\mathcal{V} = \mathrm{im}\,(\mathrm{D}\,\Phi(x_1)) = \mathrm{T}_{\Phi(x_1)}\mathcal{M}_F^{\mathrm{loc}}\,.$$

We define the following operator norms

$$\|A\| = \max_{\|w\|=1} \big|w^\top A\,w\big| = \max\{\lambda_{\max}(A),\, -\lambda_{\min}(A)\} = \sigma_{\max}(A)\,,$$

and

$$\|A|_{\mathcal{V}}\|_R = \max_{\substack{w \in \mathcal{V} \\ \|w\|=1}} \big|w^\top A\,w\big| = \max\Big\{\lambda_{\max}(A|_{\mathcal{V}}),\, -\lambda_{\min}(A|_{\mathcal{V}})\Big\} = \sigma_{\max}(A|_{\mathcal{V}})\,.$$

Also, by definition, the Lipschitz constant of $\nabla f$ is $\beta_f = \sup_{x \in \mathcal{N}} \|A\|$, and, similarly, the Lipschitz constant for the Riemannian gradient $\nabla F$ is $\beta_F = \sup_{x_1 \in \mathcal{M}_F^{\mathrm{loc}}} \|A|_{\mathcal{V}}\|_R$.

**Step 1. Restriction of the Hessian.**
By the Rayleigh quotient characterisation (see Lemma 4), we have

$$\lambda_{\max}(A|_{\mathcal{V}}) \le \lambda_{\max}(A), \qquad \lambda_{\min}(A|_{\mathcal{V}}) \ge \lambda_{\min}(A)\,,$$

so,

$$\sigma_{\max}(A|_{\mathcal{V}}) \le \sigma_{\max}(A)\,,$$

with equality if and only if there exists a unit vector $w \in \mathcal{V}$ satisfying

$$|w^\top Aw| = \sigma_{\max}(A)\,.$$

Since $\sigma_{\max}(A)$ is achieved only along vectors in the maximum singular-vector subspace

$$\Sigma_{\max} = \mathrm{span}\{v \in \mathcal{N} : Av = \lambda v,\, |\lambda| = \sigma_{\max}(A)\}\,,$$

we have equality if and only if some unit vector $w \in \mathcal{V}$ is in $\Sigma_{\max}$.

**Step 2. Uniform Non-Tangency of $\Sigma_{\max}$ with $\mathcal{V}$.**
The uniform non-tangency condition (where its general validity is expressed precisely in Corollary 6 by setting $\varepsilon = 1 - \theta$) implies that for every unit vector $v \in \Sigma_{\max}$ there exists a uniform $\varepsilon > 0$ such that
$$\|P_{\mathcal{V}}\, v\| \le 1 - \varepsilon\,.$$
Now, consider any unit vector $w \in \mathcal{V}$. Since $w$ lies in $\mathcal{V}$, we can use the following argument for any unit vector $v \in \Sigma_{\max}$:

1. Write the vector $v \in \Sigma_{\max}$ in its orthogonal decomposition with respect to $\mathcal{V}$
$$v = P_{\mathcal{V}}\, v + (I - P_{\mathcal{V}})\, v\,.$$

2. Since $w \in \mathcal{V}$, it is orthogonal to every vector in the orthogonal complement, in particular to $(I - P_{\mathcal{V}})v$. Therefore, we have,
$$w^\top v = w^\top \left[ P_{\mathcal{V}}\, v + (I - P_{\mathcal{V}})\, v \right] = w^\top P_{\mathcal{V}}\, v\,.$$

3. Applying the Cauchy-Schwartz inequality, we obtain
$$|w^\top v| = |w^\top (P_{\mathcal{V}}\, v)| \le \|w\| \|P_{\mathcal{V}}\, v\| = \|P_{\mathcal{V}}\, v\|\,.$$

   Since $\|w\| = 1$ and by the uniform non-tangency condition, it follows that
$$|w^\top v| \le 1 - \varepsilon\,.$$

**Step 3. Bounding the Absolute Rayleigh Quotient**
Let $A = \sum_{i=1}^{n} \lambda_i\, v_i v_i^\top$ be an orthonormal eigendecomposition of $A$ with eigenvectors $\{v_i\}$, and define $\sigma_i = |\lambda_i|$, ordered so that $\sigma_1 = \ldots = \sigma_p =: \sigma_{\max}(A) > \sigma_{p+1} \ge \ldots \ge \sigma_n$. For any unit vector $w \in \mathcal{V}$, write
$$w = \sum_{i=1}^{n} \alpha_i\, v_i, \qquad \text{so that} \qquad \sum \alpha_i^2 = 1\,.$$
Let $\theta := \sum_{i=2}^{p} \alpha_i^2$ be the squared norm of the projection of $w$ onto the top singular subspace. Then
$$\left| w^\top A\, w \right| = \left| \sum_{i=1}^{n} \lambda_i\, \alpha_i^2 \right| \le \sum_{i=1}^{n} \sigma_i\, \alpha_i^2 = \sigma_{\max}(A)\, \theta^2 + \sum_{i=p+1}^{n} \sigma_i \alpha_i^2\,.$$
Since $\sigma_i \le \sigma_{p+1}$ for $i > p$, the tail sum is bounded:
$$\sum_{i>p} \sigma_i \alpha_i^2 \le \sigma_{p+1}(1 - \theta^2)\,,$$
yielding
$$\left| w^\top A\, w \right| \le \sigma_{\max}(A)\, \theta^2 + \sigma_{p+i}(1 - \theta^2)\,.$$
Using the uniform non-tangency condition $\theta^2 \le (1 - \varepsilon)^2$ and taking the maximum over all $w \in \mathcal{V}$, we obtain
$$\sigma_{\max}(A|_{\mathcal{V}}) = \max_{w \in \mathcal{V},\, \|w\|=1} \left| w^\top A\, w \right| \le \sigma_{\max}(A)\, (1 - \varepsilon)^2 + \sigma_{p+1} \left[ 1 - (1 - \varepsilon)^2 \right]\,.$$

**Step 4. Expressing the Gap via the Singular-Value Gap**
We define the spectral gap between the largest eigenvalue and the second largest as
$$\Delta_{\max} \le \sigma_{\max}(A) - \sigma_{p+1}\,.$$
Rewriting $\sigma_{p+1} \le \sigma_{\max}(A) - \Delta_{\max}$, we have,
$$\begin{aligned}
\sigma_{\max}(A|_{\mathcal{V}}) &\le \sigma_{\max}(A)\, (1 - \varepsilon)^2 + (\sigma_{\max}(A) - \Delta_{\max}) \left[ 1 - (1 - \varepsilon)^2 \right] \\
&= \sigma_{\max}(A) - \Delta_{\max} \left[ 1 - (1 - \varepsilon)^2 \right] \\
&= \sigma_{\max}(A) - \Delta_{\max}(2\varepsilon - \varepsilon^2)\,.
\end{aligned}$$

Taking the supremum over all points in each domain, it follows that

$$\beta_F = \sup_{x_1 \in \mathcal{M}_F^{\mathrm{loc}}} \sigma_{\max}(A|_{\mathcal{V}}) \le \sup_{x \in \mathcal{N}} \sigma_{\max}(A) - \Delta_{\max}(2\varepsilon - \varepsilon^2) = \beta_f - \Delta_{\max}(2\varepsilon - \varepsilon^2).$$

Thus, by our assumption that $\Delta_{\max} > 0$, we obtain the desired strict inequality:

$$\beta_F \le \beta_f - \Delta_{\max}(2\varepsilon - \varepsilon^2) < \beta_f.$$

This completes the proof. ∎

## C.2 Proof of Corollary 1

**Proof** The result follows directly from Corollary 4. Applying this to $\nabla^2 F(x_1)$ and take the supremum over $x_1 \in \mathcal{M}_F^{\mathrm{loc}}$ yields

$$\beta_F^{(E)} = \sup_{x_1} \big\| \nabla^2 F(x_1) \big\| \le M^{(\Phi)} \sup_{x_1} \big\| \nabla^2 F(x_1) \big\|_R = M^{(\Phi)} \beta_F.$$

Finally, substitute the bound $\beta_F \le \beta_f - \Delta_{\max}(2\varepsilon - \varepsilon^2)$ from Theorem 1 to conclude

$$\beta_F^{(E)} \le M^{(\Phi)} \big[ \beta_f - \Delta_{\max}(2\varepsilon - \varepsilon^2) \big] < M^{(\Phi)} \beta_f,$$

with $M^{(\Phi)}$ be equal to 2 by the non-expansive property of the projection mapping $\Psi$. ∎

## C.3 Proof of Theorem 2

**Proof** From Lemma 3, we have that the Hessian of $F$ can be written as

$$\nabla^2 F(x_1) = \mathrm{D}\,\Phi(x_1)^\top \nabla^2 f\big(\Phi(x_1)\big)\,\mathrm{D}\,\Phi(x_1) + \mathrm{C}(x_1),$$

where the correction term due to the nonlinearity of $\Psi(x_1)$ is

$$\mathrm{C}(x_1) = \mathrm{D}^2\,\Psi(x_1) \bullet \nabla_{x_2} f\big(\Phi(x_1)\big).$$

Here $\bullet$ denotes the contraction of the third-order Hessian tensor $\mathrm{D}^2\,\Psi \in \mathbb{R}^{n_2 \times n_1 \times n_1}$ with the gradient vector $\nabla_{x_2} f \in \mathbb{R}^{n_2}$ over the first index, that is,

$$\mathrm{C}(x_1) = \sum_{k=1}^{n_2} \mathrm{D}^2\,\Psi_{k,:,:}(x_1) \cdot \frac{\partial f}{\partial x_{2,k}}\big(\Phi(x_1)\big).$$

Define $A := \nabla^2 f(\Phi(x_1))$, $B := \mathrm{D}\,\Phi(x_1)^\top \nabla^2 f\big(\Phi(x_1)\big)\,\mathrm{D}\,\Phi(x_1)$, and the restriction of $A$ into the tangent space of $\mathcal{V} = \mathrm{T}_{\Phi(x_1)}\mathcal{M}_F^{\mathrm{loc}}$ as $A|_{\mathcal{V}} = \nabla^2 F(x_1)$, such that

$$A|_{\mathcal{V}} = B + \mathrm{C}.$$

Since $A|_{\mathcal{V}}$ and $B$ are symmetric, it follows that $\mathrm{C}(x_1)$ is a symmetric matrix. For $B, \mathrm{C}$ symmetric matrices, we apply the Weyl's inequality for the pullback metric $R$

$$\sigma_{\max}(B + \mathrm{C}) \le \sigma_{\max}(B) + \|\mathrm{C}\|_R.$$

From Theorem 1, we have

$$\sigma_{\max}(B) \le \sigma_{\max}(A) - \Delta_{\max}\left(2\varepsilon - \varepsilon^2\right),$$

so,

$$\sigma_{\max}(A|_{\mathcal{V}}) \le \sigma_{\max}(A) - \Delta_{\max}\left(2\varepsilon - \varepsilon^2\right) + \|\mathrm{C}\|_R,$$

where we have $\|C\|_R \le \|C\|/m^{(\Phi)}$. We need to control now the Euclidean operator norm of the correction term C. By the sub-multiplicative property of the norm and our assumption on the bounds, we get

$$\|\mathrm{C}\| \le \|\mathrm{D}^2\,\Psi(x_1)\|\,\|\nabla_{x_2} f(\Phi(x_1))\| \le Q\,Z,$$

where since $\mathrm{D}^2\,\Psi(x_1)$ is a higher order tensor, we take the appropriate norm for the sub-multiplicative property to hold. More specifically, a standard choice is to use the tensor operator or injective norm. If we have an $N$-th order tensor $\mathsf{T} \in \mathbb{R}^{n_1 \times n_2 \times \cdots \times n_N}$ its operator norm is defined by

$$\|\mathsf{T}\| = \sup\{|\mathsf{T}(x^{(1)}, x^{(2)}, \ldots, x^{(N)})| : \|x^{(i)}\| = 1,\ \forall i\},$$

where $\mathbf{T}(x^{(1)}, x^{(2)}, \ldots, x^{(N)})$ is the scalar obtained by contracting $\mathbf{T}$ with the vectors $x^{(1)}, \ldots, x^{(N)}$.

We also have the uniform curvature gap defined as

$$\delta = \sigma_{\max}(A) - \sigma_{\max}(B) \geq \eta \,.$$

Since the bound $Q\,Z$ on the correction term is assumed to be strictly less than the curvature gap $\delta$, the improvement in smoothness obtained by projecting onto the tangent space is preserved. In other words, the additional curvature introduced by the nonlinearity does not overwhelm the inherent reduction from the projection, ensuring that the overall smoothness constant of the reduced function remains strictly below $\beta_f$.

Formally, by taking the supremum over the region of interest, the Lipschitz constant $\beta_F$ of the Riemannian gradient $\nabla F$ satisfies

$$\beta_F \leq \beta_f - \Delta_{\max}(2\varepsilon - \varepsilon^2) + \frac{Q\,Z}{m^{(\Phi)}} < \beta_f \,.$$

For projection mappings $\Psi$, we have $0 \preceq \mathrm{D}\,\Psi^\top \mathrm{D}\,\Psi \preceq I$, and $I \preceq \mathrm{D}\,\Phi^\top \mathrm{D}\,\Phi \preceq 2I$. Hence $m^{(\Phi)} = 1$. This completes the proof. ∎

# D  Additional Theoretical Results

This appendix presents supplementary theoretical results that support the main developments in the paper. While not required for the core proofs, these results offer deeper insight into the structure of reduction mappings and their implications for optimisation and geometric analysis.

Specifically:

- Section D.1 provides a quantitative bound on the correction term induced by composing a function $f$ with a smooth minimising mapping $\Psi$, derived via the implicit function theorem.
- Section D.2 establishes that Morse–Bott structure is preserved under reduction, assuming mild regularity and a clean intersection condition.

## D.1  Correction Term Bound Under Argmin Mappings

The proof of the main result below relies on expressions for the first- and second-order derivatives of implicit argmin mappings. These are provided in Appendix E, Subsection E.1.

**Theorem 4 (Quantitative Bound on the Correction Term)** *Let* $f : \mathbb{R}^n \to \mathbb{R}$ *be* $C^2$*, and let* $G : \mathbb{R}^{n_1} \times \mathbb{R}^{n_2} \to \mathbb{R}$ *be* $C^3$*. Assume that for each* $x_1$ *in an open neighbourhood* $\mathcal{U} \subset \mathbb{R}^{n_1}$*, the solution mapping*

$$\Psi(x_1) = \underset{u \in \mathbb{R}^{n_2}}{\arg\min}\ G(x_1, u) \,,$$

*is well-defined and* $C^2$*. Suppose further that for every* $x_1 \in \mathcal{U}$*:*

*(i)* **Strict second-order sufficient condition (SSOSC).**

$$H(x_1) := \nabla_u^2 G(x_1, \Psi(x_1)) \succeq \sigma I_{n_2}, \quad \sigma > 0 \,.$$

*(ii)* **Bounded higher-order derivatives.** *There exist constants* $L_{12}, L_{21}, L_3, L_{11} > 0$ *such that at* $(x_1, \Psi(x_1))$*,*

$$\left\| \nabla_{x_1^2 u}^3 G \right\| \leq L_{12}, \quad \left\| \nabla_{x_1 u^2}^3 G \right\| \leq L_{21}, \quad \left\| \nabla_{u^3}^3 G \right\| \leq L_3, \quad \left\| \nabla_{x_1 u}^2 G \right\| \leq L_{11}.$$

*Write*

$$\nabla f = (\nabla_{x_1} f, \nabla_{x_2} f), \quad v(x_1) = \nabla_{x_2} f(x_1, \Psi(x_1)), \quad \|\nabla f(x_1, \Psi(x_1))\| \leq L_f.$$

*Define*

$$\xi = \begin{cases} \dfrac{\|v(x_1)\|}{\|\nabla f(x_1, \Psi(x_1))\|}, & \|\nabla f(x_1, \Psi(x_1))\| > 0, \\ 0, & \textit{otherwise}, \end{cases} \qquad \xi \in [0, 1] \,.$$

*Let*

$$\mathrm{D}^2\,\Psi(x_1)\colon \mathbb{R}^{n_1}\times\mathbb{R}^{n_1}\to\mathbb{R}^{n_2}, \quad \mathrm{C}(x_1)\colon \mathbb{R}^{n_1}\times\mathbb{R}^{n_1}\to\mathbb{R},$$

*be defined by*

$$\mathrm{C}(x_1)(h,k)=\Big\langle v(x_1),\ \mathrm{D}^2\,\Psi(x_1)[h,k]\Big\rangle.$$

*Define the operator norms*

$$\|\,\mathrm{D}^2\,\Psi(x_1)\|=\sup_{\|h\|=\|k\|=1}\big\|\mathrm{D}^2\,\Psi(x_1)[h,k]\big\|, \quad \|\mathrm{C}(x_1)\|=\sup_{\|h\|=\|k\|=1}\big|\mathrm{C}(x_1)(h,k)\big|,$$

*and set*

$$\tilde{L}=\sigma\,L_{12}+L_{21}\,L_{11}+\frac{L_3\,L_{11}^2}{\sigma}\,.$$

*Then for every $x_1\in\mathcal{U}$, the following bounds hold:*

$$\|v(x_1)\|\le\xi\,L_f,\quad \|\,\mathrm{D}^2\,\Psi(x_1)\|\le\frac{\tilde{L}}{\sigma^2},\quad \|\mathrm{C}(x_1)\|\le\frac{\tilde{L}}{\sigma^2}\,\xi\,L_f\,.$$

*Moreover, if one regards $\mathrm{D}^2\,\Psi(x_1)$ as a linear map $\mathrm{Sym}^2(\mathbb{R}^{n_1})\to\mathbb{R}^{n_2}$ with top singular subspace $\Sigma\subset\mathbb{R}^{n_2}$ and defines $\cos\theta=\|\mathrm{P}_\Sigma\,v(x_1)\|/\|v(x_1)\|$, then the refined estimate*

$$\|\mathrm{C}(x_1)\|\le\frac{\tilde{L}}{\sigma^2}\,\|v(x_1)\|\,\cos\theta\le\frac{\tilde{L}}{\sigma^2}\,\xi\,L_f\,\cos\theta$$

*also holds.*

**Proof**   We assume the mapping arises from an unconstrained problem for simplicity; analogous results hold in the constrained setting.

**1. First derivative via IFT.** By the SSOSC assumption,

$$H=\nabla_u^2 G(x_1,\Psi(x_1))\succeq\sigma I\ \Rightarrow\ \|H^{-1}\|\le\frac{1}{\sigma}\,.$$

Applying the first-order implicit function theorem (Lemma 1) yields

$$\mathrm{D}\,\Psi(x_1)=-H^{-1}\nabla_{x_1 u}^2 G,\quad \|\,\mathrm{D}\,\Psi\|\le\frac{L_{11}}{\sigma}\,.$$

**2. Second derivative via higher-order IFT.** Let

$$\mathbf{A}=\nabla_{x_1^2 u}^3 G,\quad \mathbf{B}=\nabla_{x_1 u^2}^3 G,\quad \mathbf{C}=\nabla_{u^3}^3 G\,.$$

Then the second-order expansion (Lemma 2) gives

$$\mathrm{D}^2\,\Psi=-H^{-1}\Big[\mathbf{A}+\mathscr{S}\big[\mathbf{B}\bullet(I\otimes\mathrm{D}\,\Psi)\big]+\mathbf{C}\bullet(\mathrm{D}\,\Psi\otimes\mathrm{D}\,\Psi)\Big]\,.$$

Using the bounds

$$\|H^{-1}\|\le\tfrac{1}{\sigma},\quad \|\,\mathrm{D}\,\Psi\|\le\tfrac{L_{11}}{\sigma},\quad \|\mathbf{A}\|\le L_{12},\quad \|\mathbf{B}\|\le L_{21},\quad \|\mathbf{C}\|\le L_3\,,$$

we obtain

$$\|\,\mathrm{D}^2\,\Psi\|\le\frac{1}{\sigma}\left(L_{12}+L_{21}\frac{L_{11}}{\sigma}+L_3\frac{L_{11}^2}{\sigma^2}\right)=\frac{\tilde{L}}{\sigma^2}\,.$$

**3. Partial-gradient bound.** Since $\|v\|\le\|\nabla f\|\le L_f$, we have

$$\|v\|=\xi\,\|\nabla f\|\le\xi\,L_f\,.$$

Moreover, $\xi<1$ strictly whenever $\nabla_{x_1} f(x_1,\Psi(x_1))\ne 0$, since then

$$\|\nabla f\|>\|\nabla_{x_2} f\|=\|v\|\,.$$

The degenerate case $\xi=1$ occurs only if $\nabla_{x_1} f\equiv 0$, i.e., the full gradient lacks an $x_1$-component.

**4. Operator norm of the correction form.** By Cauchy–Schwarz,

$$\|C\| = \sup_{\|h\|=\|k\|=1} \left| \langle v, \, D^2 \, \Psi[h,k] \rangle \right| \leq \|v\| \cdot \| \, D^2 \, \Psi \| \leq \frac{\tilde{L}}{\sigma^2} \, \xi \, L_f \, .$$

Let $\Sigma$ denote the top singular subspace of $D^2 \, \Psi$. Projecting $v$ onto $\Sigma$, we obtain the refined bound

$$\|C\| \leq \| \, D^2 \, \Psi \| \cdot \|v\| \cdot \cos\theta \, ,$$

where $\cos\theta = \|P_\Sigma v\| / \|v\|$.

Equality $\cos\theta = 1$ holds only when $v \in \Sigma$, i.e., when $v$ is fully aligned with the top-amplification directions. Otherwise, $\cos\theta < 1$ strictly whenever $v(x_1) \notin \Sigma$. ∎

### D.2 Preservation of Morse–Bott Structure under Reduction Mappings

**Assumption 3** *The intersection $\mathcal{M}_F \cap \mathcal{S}$ is clean[2], ensuring $\mathcal{S}_F$ is a $C^1$ submanifold.*

**Remark 4 (Correspondence of critical sets under reduction mappings)** *Let $\Phi(x_1) := (x_1, \Psi(x_1))$ and $F(x_1) := f(\Phi(x_1))$. Then $\Phi$ is a global $C^2$-diffeomorphism onto its graph manifold $\mathcal{M}_F$. If the critical set $\mathcal{S}$ intersects $\mathcal{M}_F$ cleanly, their intersection corresponds diffeomorphically to the reduced critical set $\mathcal{S}_F = \{x_1 : F(x_1) = c, \nabla F(x_1) = 0\}$, with*

$$\Phi : \mathcal{S}_F \xrightarrow{\cong} \mathcal{S} \cap \mathcal{M}_F \, .$$

*Moreover, $D \, \Phi(x_1)$ is a linear isomorphism between their tangent spaces.*

**Theorem 5 (Morse–Bott Reduction)** *Let $f : \mathbb{R}^n \to \mathbb{R}$ be a $C^2$ function which, on a compact neighbourhood $\mathcal{N}$ of a level $c$, satisfies the Morse–Bott property: its critical set $\mathcal{S} = \{x \in \mathcal{N} : \nabla f(x) = 0, \, f(x) = c\}$, is a smooth submanifold and*

$$\ker\left(\nabla^2 f(x)\right) = T_x \mathcal{S} \qquad \forall x \in \mathcal{S} \, ,$$

*while every positive eigenvalue of $\nabla^2 f(x)$ satisfies $\lambda_i \geq \mu_f > 0$ uniformly.*

*Let $\Phi : \mathbb{R}^{n_1} \to \mathbb{R}^{n_1+n_2}$ be a $C^2$ embedding and denote its image $\mathcal{M}_F = \Phi(\mathbb{R}^{n_1})$. Write $\Phi(x_1) = (x_1, \Psi(x_1))$, so $\Psi : \mathbb{R}^{n_1} \to \mathbb{R}^{n_2}$ is the induced projection. Define the reduced function*

$$F : \mathbb{R}^{n_1} \to \mathbb{R}, \qquad F(x_1) = f\left(\Phi(x_1)\right) \, ,$$

*and set*

$$\mathcal{N}_F = \Phi^{-1}(\mathcal{N}), \qquad \mathcal{S}_F = \left\{x_1 \in \mathcal{N}_F : \nabla F(x_1) = 0, \, F(x_1) = c\right\} \, .$$

*Also suppose that for the intersection $\mathcal{S} \cap \mathcal{M}_F$ Assumptions 2 and 3 hold. The $F$ is Morse–Bott on $\mathcal{N}_F$:*

1. *$\mathcal{S}_F$ is a smooth submanifold of $\mathbb{R}^{n_1}$.*

2. *At each $x_1 \in \mathcal{S}_F$, $\ker\left(\nabla^2 F(x_1)\right) = T_{x_1} \mathcal{S}_F$, and every positive eigenvalue of $\nabla^2 F(x_1)$ satisfies $\lambda_i \geq \mu_F > 0$ uniformly.*

**Proof** Since, by assumption, the submanifolds $\mathcal{S}$ and $\mathcal{M}_F$ intersect cleanly, their intersection $\mathcal{S} \cap \mathcal{M}_F$ is a smooth submanifold of $\mathbb{R}^n$. Moreover, since $\Phi$ is a $C^2$ embedding, the pre-image $\mathcal{S}_F = \Phi^{-1}(\mathcal{S} \cap \mathcal{M}_F)$ is a smooth submanifold of $\mathbb{R}^{n_1}$, diffeomorphic to $\mathcal{S} \cap \mathcal{M}_F$ (see Remark 4).

At a critical point $x_1^* \in \mathcal{S}_F$, we now establish the identity

$$\ker\left(\nabla^2 F(x_1^*)\right) = T_{x_1^*} \mathcal{S}_F \, .$$

The proof consists of two parts.

---

[2]We assume that the intersection $\mathcal{M}_F \cap \mathcal{S}$ is clean so that $\mathcal{S}_F$ is a well-defined smooth submanifold. This assumption is essential for establishing the Morse–Bott property for the reduced function $F$. A discussion on the definition of clean intersections and their genericity is provided in Appendix F.

**(1)** $\mathrm{T}_{x_1^*}\mathcal{S}_F \subset \ker\big(\nabla^2 F(x_1^*)\big)$**:**

A tangent $w \in \mathrm{T}_{x_1^*}\mathcal{S}_F$ must satisfy $\nabla F = 0$, $F = c$ to first order. At a critical point $\nabla F(x_1^*) = 0$, the only nontrivial condition is

$$\nabla^2 F(x_1^*)\, w = 0\,,$$

so that $w \in \ker\big(\nabla^2 F(x_1^*)\big)$.

**(2)** $\ker\big(\nabla^2 F(x_1^*)\big) \subset \mathrm{T}_{x_1^*}\mathcal{S}_F$**:**

If $\nabla^2 F(x_1^*)\, w = 0$, then

$$0 = \mathrm{D}\,\Phi(x_1^*)^\top \left[\nabla^2 f\big(\Phi(x_1^*)\big)\, \mathrm{D}\,\Phi(x_1^*)\, w\right]\,.$$

Injectivity of $\mathrm{D}\,\Phi$ gives $\nabla^2 f\big(\Phi(x_1^*)\big)\,[\mathrm{D}\,\Phi(x_1^*)\, w] = 0$, so

$$\mathrm{D}\,\Phi(x_1^*)\, w \in \ker\left(\nabla^2 f\big(\Phi(x_1^*)\big)\right)\,.$$

Since $f$ satisfies the Morse–Bott property, we have

$$\ker\left(\nabla^2 f\big(\Phi(x_1^*)\big)\right) = \mathrm{T}_{\Phi(x_1^*)}\mathcal{S}\,.$$

By definition, we have $\mathrm{D}\,\Phi(x_1^*)\, w \in \mathrm{T}_{\Phi(x_1^*)}\mathcal{M}_F$, hence

$$\mathrm{D}\,\Phi(x_1^*)\, w \in \mathrm{T}_{\Phi(x_1^*)}\mathcal{S} \cap \mathrm{T}_{\Phi(x_1^*)}\mathcal{M}_F = \mathrm{T}_{\Phi(x_1^*)}(\mathcal{S} \cap \mathcal{M}_F)\,.$$

By the pre-image description of $\mathcal{S}_F$, we conclude that $w$ must lie in $\mathrm{T}_{x_1^*}\mathcal{S}_F$.

Combining the two inclusions yields

$$\ker\big(\nabla^2 F(x_1^*)\big) = \mathrm{T}_{x_1^*}\mathcal{S}_F\,,$$

which shows that the reduced function $F(x_1)$ satisfies the Morse–Bott property. On a compact $\mathcal{S}_F$, the smallest positive eigenvalue of $\nabla^2 F$ attains a minimum of $\mu_F > 0$. ∎

## E    Supporting Lemmas

This appendix collects technical results used throughout the main body of the paper. The lemmas are grouped into four self-contained subsections, each addressing a specific analytical or geometric aspect of the theory:

- **Derivatives:** Closed-form expressions for the first- and second-order derivatives of implicitly defined mappings, as well as the Hessian of the reduced function induced by a smooth reparametrisation.
- **Rayleigh Quotients Comparison:** Classical inequalities comparing spectral quantities of symmetric matrices under embeddings into lower-dimensional subspaces, used to control the curvature of reduced objectives.
- **Uniform Angle Bounds and Genericity:** Results showing that the eigenspaces associated with extreme curvature directions intersect the tangent space of the feasible manifold only trivially. This condition holds generically and yields uniform lower bounds on the angle between the subspaces.
- **Genericity of Single Eigenvalues:** A measure-theoretic argument demonstrating that symmetric matrices with repeated eigenvalues form a zero-measure subset. This justifies the generic spectral gap assumptions used in several proofs.

### E.1    Derivatives

**Lemma 1 (First-Order Derivative Expression for Implicit Functions)**  *Let $h : \mathbb{R}^m \times \mathbb{R}^n \to \mathbb{R}$ be at least twice continuously differentiable. Suppose that for each $x$ there exists a unique $y^*(x)$ satisfying the optimality condition*

$$\mathrm{D}_y\, h(x, y^*(x)) = 0_{1\times n}\,,$$

*and assume further that the Hessian* $\mathrm{D}_y^2\, h(x, y^*(x))$ *is invertible for all* $x$ *in a neighbourhood* $\mathcal{U}$ *around a local minimiser. Then the first-order derivative of* $y^*(x)$ *with respect to* $x$ *is given by*

$$\mathrm{D}\, y^*(x) = -\left[\mathrm{D}_y^2\, h(x, y^*(x))\right]^{-1} \mathrm{D}_{xy}^2\, h(x, y^*(x))\,.$$

**Proof**    Starting from the optimality condition and differentiating it with respect to $x$ yields

$$\mathrm{D}_x(\mathrm{D}_y\, h(x, y^*(x)))^\top = 0$$

$$\mathrm{D}_{xy}^2\, h(x, y^*(x)) + \mathrm{D}_y^2\, h(x, y^*(x))\, \mathrm{D}\, y^*(x) = 0$$

$$\mathrm{D}\, y^*(x) = -\left[\mathrm{D}_{yy}^2\, h(x, y^*(x))\right]^{-1} \mathrm{D}_{xy}^2\, h(x, y^*(x))\,.$$

The uniqueness of $y^*(x)$ is guaranteed by the classical implicit function theorem.    ∎

**Lemma 2 (Second-Order Derivative Expression for Implicit Functions)**    *Let* $h : \mathbb{R}^m \times \mathbb{R}^n \to \mathbb{R}$ *be* $C^3$*, and suppose that* $y^*(x)$ *is the unique solution of*

$$\mathrm{D}_y\, h(x, y^*(x)) = 0, \quad x \in \mathcal{U}\,,$$

*with*

$$H(x) := \mathrm{D}_y^2\, h(x, y^*(x)) \in \mathrm{GL}_n(\mathbb{R})\,,$$

*for all* $x \in \mathcal{U}$*. Define*

$$J(x) := \mathrm{D}\, y^*(x) = -H(x)^{-1}\, \mathrm{D}_{xy}^2\, h\, (x, y^*(x)) \in \mathbb{R}^{n \times m}\,.$$

*Further introduce the following third-order multilinear maps at* $(x, y^*(x))$*:*

$$\mathbf{A}(x) := \mathrm{D}_{x^2 y}^3\, h \in \mathbb{R}^n \otimes \mathbb{R}^m \otimes \mathbb{R}^m,$$

$$\mathbf{B}(x) := \mathrm{D}_{xy^2}^3\, h \in \mathbb{R}^n \otimes \mathbb{R}^m \otimes \mathbb{R}^n,$$

$$\mathbf{C}(x) := \mathrm{D}_y^3\, h \in \mathbb{R}^n \otimes \mathbb{R}^n \otimes \mathbb{R}^n\,.$$

*Let* $\mathscr{S} : L(\mathbb{R}^m \otimes \mathbb{R}^m, \mathbb{R}^n) \to L(\mathrm{Sym}^2\mathbb{R}^m, \mathbb{R}^n)$ *be the symmetrisation operator* $\mathscr{S}[\mathbf{T}](u, v) = \frac{1}{2}(\mathbf{T}[u, v] + \mathbf{T}[v, u])$*. Then the second derivative* $\mathrm{D}^2\, y^*(x) \in L(\mathrm{Sym}^2\mathbb{R}^m, \mathbb{R}^n)$ *is given by the formula*

$$\mathrm{D}^2\, y^*(x) = -H(x)^{-1} \bullet [\mathbf{A}(x) + \mathscr{S}[\mathbf{B}(x) \bullet (I \otimes J(x))] + \mathbf{C}(x) \bullet (J(x) \otimes J(x))]\,,$$

*where* $\otimes$ *is the tensor product and* $\bullet$ *is a tensor contraction on matching covariant and contravariant indices.*

**Proof**    For convenience, we define the vector-valued function $F : \mathbb{R}^m \times \mathbb{R}^n \to \mathbb{R}^n$ where

$$F(x, y) = \mathrm{D}_y\, h(x, y), \quad \mathrm{D}_x\, F = \mathrm{D}_{xy}^2\, h, \quad \mathrm{D}_y\, F = \mathrm{D}_{y^2}^2\, h = H(x)\,.$$

From Lemma 1, we have that for the direction $u \in \mathbb{R}^m$

$$\mathrm{D}\, F[u] = \mathrm{D}_x\, F[u] + \mathrm{D}_y\, F\big[\mathrm{D}\, y^*[u]\big] = 0\,.$$

Differentiating again in direction $v \in \mathbb{R}^m$, we obtain

$$\mathrm{D}\, (\mathrm{D}_x\, F[u])\, [v] + \mathrm{D}\, \big(\mathrm{D}_y\, F\big[\mathrm{D}\, y^*[u]\big]\big)\, [v] = 0\,.$$

We now expand each of these two terms by applying the produce - and chain - rules in turn.

$$\mathrm{D}\, (\mathrm{D}_x\, F[u])\, [v] = \underbrace{\mathrm{D}_{xx}^2\, F[u, v]}_{\mathbf{A}(x)[u, v]} + \underbrace{\mathrm{D}_{xy}^2\, F[u, \mathrm{D}\, y^*[v]]}_{\mathbf{B}(x)[u, \mathrm{D}\, y^*[v]]}\,.$$

$$\mathrm{D}\, \big(\mathrm{D}_y\, F\big[\mathrm{D}\, y^*[u]\big]\big)\, [v] = \mathbf{B}(x)[\mathrm{D}\, y^*[u], v] + \underbrace{\mathrm{D}_{yy}^2\, F[\mathrm{D}\, y^*[u], \mathrm{D}\, y^*[v]]}_{\mathbf{C}(x)[\mathrm{D}\, y^*[u], \mathrm{D}\, y^*[v]]} + \mathrm{D}_y\, F[\mathrm{D}^2\, y^*[u, v]]\,.$$

Summing the two expansions and recalling that $\mathrm{D}_y\, F = \mathrm{D}_y^2\, h$ is the Hessian in $y$, we get

$$\mathrm{D}_y^2\, h[\mathrm{D}^2\, y^*[u, v]] = -\big\{\mathbf{A}(x)[u, v] + \mathbf{B}(x)[u, \mathrm{D}\, y^*[v]] + \mathbf{B}(x)[\mathrm{D}\, y^*[u], v] + \mathbf{C}(x)[\mathrm{D}\, y^*[u], \mathrm{D}\, y^*[v]]\big\}\,.$$

Since $D_y^2 h =: H(x)$ is invertible, and denoting $J(x) := D\, y^*(x)$, we conclude

$$D^2\, y^*(x)[u, v] = -[H(x)]^{-1}\big\{\mathbf{A}(x)[u, v] + \mathbf{B}(x)[u, J(v)] + \mathbf{B}(x)[J(u), v] + \mathbf{C}(x)[J(u), J(v)]\big\}\,.$$

Because the function $h$ is smooth, the mixed partials should commute by the Schwarz's theorem. The tensors $\mathbf{A}$ and $\mathbf{C}$ are already symmetric, so we only need to deal with the two $\mathbf{B}$ terms. To that end, we introduce the symmetrisation operator $\mathscr{S}[\mathbf{T}](u, v) = \frac{1}{2}(\mathbf{T}[u, v], \mathbf{T}[v, u])$, and we conclude

$$D^2\, y^*(x)[u, v] = -[H(x)]^{-1}\big\{\mathbf{A}(x)[u, v] + \mathscr{S}[\mathbf{B}(\cdot, J(\cdot))](u, v) + \mathbf{C}(x)[J(u), J(v)]\big\}\,.$$

Re-expressing this result in the compact tensor-product/contraction notation exactly yields the lemma's statement. ∎

**Lemma 3 (Restricted Hessian of the Reduced Function)** *Let $f\colon \mathbb{R}^{n_1 + n_2} \to \mathbb{R}$ be twice continuously differentiable and let $\Psi(x_1)\colon \mathbb{R}^{n_1} \to \mathbb{R}^{n_2}$ be a twice continuously differentiable mapping. Define the reduced function $F\colon \mathbb{R}^{n_1} \to \mathbb{R}$ by $F(x_1) = f\big(x_1, \Psi(x_1)\big)$, and denote the reduction map*

$$\Phi(x_1) = \begin{pmatrix} x_1 \\ \Psi(x_1) \end{pmatrix}, \quad \text{with} \quad J(x_1) = D_{x_1}\, \Psi(x_1)\,.$$

*Then the Hessian of $F$ is given by the compact representation*

$$\nabla^2 F(x_1) = \begin{bmatrix} I & J(x_1)^\top \end{bmatrix} \nabla^2 f(x^*) \begin{bmatrix} I \\ J(x_1) \end{bmatrix} + \mathrm{C}(x_1)\,,$$

*where $x^* = \Phi(x_1)$ and the correction term*

$$\mathrm{C}(x_1) = D_{x_1}^2\, \Psi(x_1) \bullet \nabla_{x_2} f(x^*)\,,$$

*captures the curvature of the constraint manifold $\mathcal{M}_F = \{(x_1, \Psi(x_1))\colon x_1 \in \mathbb{R}^{n_1}\}$. The symbol $\bullet$ denotes a tensor contraction operation at the appropriate indices.*

**Proof** We start by differentiating the reduced function $F(x_1) = f(x_1, \Psi(x_1))$. By the chain rule,

$$\nabla F(x_1) = \nabla_{x_1} f(x^*) + (D_{x_1}\, \Psi(x_1))^\top \nabla_{x_2} f(x^*)\,,$$

where $x^* = (x_1, \Psi(x_1))$. Differentiating $\nabla F(x_1)$ with respect to $x_1$ gives

$$\nabla^2 F(x_1) = D_{x_1}\Big(\nabla_{x_1} f(x^*) + (D_{x_1}\, \Psi(x_1))^\top \nabla_{x_2} f(x^*)\Big)$$

$$= \nabla_{x_1}^2 f(x^*) + \nabla_{x_1 x_2}^2 f(x^*)\, D_{x_1}\, \Psi(x_1)$$

$$+ D_{x_1}\Big[(D_{x_1}\, \Psi(x_1))^\top\Big] \bullet \nabla_{x_2} f(x^*) + (D_{x_1}\, \Psi(x_1))^\top D_{x_1}\Big[\nabla_{x_2} f(x^*)\Big]\,.$$

Since $D_{x_1}[\nabla_{x_2} f(x^*)] = \nabla_{x_2 x_1}^2 f(x^*) + \nabla_{x_2}^2 f(x^*)\, D_{x_1}\, \Psi(x_1)$ and using the notation $J(x_1) = D_{x_1}\, \Psi(x_1)$, we can rewrite the above as

$$\nabla^2 F(x_1) = \nabla_{x_1}^2 f(x^*) + \nabla_{x_1 x_2}^2 f(x^*)\, J(x_1)$$

$$+ D_{x_1}\Big[(J(x_1))^\top\Big] \bullet \nabla_{x_2} f(x^*) + J(x_1)^\top\Big[\nabla_{x_2 x_1}^2 f(x^*) + \nabla_{x_2}^2 f(x^*)\, J(x_1)\Big]\,.$$

Grouping like terms, we obtain

$$\nabla^2 F(x_1) = \nabla_{x_1}^2 f(x^*) + \nabla_{x_1 x_2}^2 f(x^*)\, J(x_1) + J(x_1)^\top \nabla_{x_2 x_1}^2 f(x^*) + J(x_1)^\top \nabla_{x_2}^2 f(x^*)\, J(x_1)$$

$$+ \underbrace{\underbrace{D_{x_1}^2\, \Psi(x_1)}_{n_2 \times n_1 \times n_1} \bullet \underbrace{\nabla_{x_2} f(x^*)}_{n_2 \times 1}}_{n_1 \times n_1}\,.$$

This expression can be compactly represented in block form as

$$\nabla^2 F(x_1) = \begin{bmatrix} I & J(x_1)^\top \end{bmatrix} \nabla^2 f(x^*) \begin{bmatrix} I \\ J(x_1) \end{bmatrix} + D_{x_1}^2\, \Psi(x_1) \bullet \nabla_{x_2} f(x^*)\,,$$

which completes the derivation. ∎

### E.2 Rayleigh Quotients Comparison

**Lemma 4 (Rayleigh Quotient on an Embedded Subspace)** *Let $A \in \mathbb{R}^{n \times n}$ be a symmetric matrix and let $B \in \mathbb{R}^{n \times k}$ ($k < n$) have full column–rank. Write*

$$\mathcal{V} := \operatorname{im} B \subset \mathbb{R}^n, \qquad B^\top B \succ 0 \,.$$

*Define the ambient extremal Rayleigh quotients*

$$\lambda_{\max}(A) := \max_{v \in \mathbb{R}^n, \, v \neq 0} \frac{v^\top A v}{v^\top v}, \qquad \lambda_{\min}(A) := \min_{v \in \mathbb{R}^n, \, v \neq 0} \frac{v^\top A v}{v^\top v},$$

*and the embedded (or generalised) Rayleigh quotients*

$$\lambda_{\max}(A|_{\mathcal{V}}) := \max_{y \in \mathbb{R}^k, \, y \neq 0} \frac{y^\top B^\top A B \, y}{y^\top B^\top B \, y}, \qquad \lambda_{\min}(A|_{\mathcal{V}}) := \min_{y \in \mathbb{R}^k, \, y \neq 0} \frac{y^\top B^\top A B \, y}{y^\top B^\top B \, y} \,.$$

*Then*

$$\lambda_{\min}(A) \leq \lambda_{\min}(A|_{\mathcal{V}}) \leq \lambda_{\max}(A|_{\mathcal{V}}) \leq \lambda_{\max}(A) \,. \tag{$*$}$$

*Moreover, the right (resp. left) inequality in $(*)$ is strict if and only if $E_{\max}(A) \cap \mathcal{V} = \{0\}$ (resp. $E_{\min}(A) \cap \mathcal{V} = \{0\}$).*

**Proof**  Put $w := By \in \mathcal{V}$. Because $B$ has full column–rank, the map $y \mapsto w$ is a bijection between $\mathbb{R}^k \setminus \{0\}$ and $\mathcal{V} \setminus \{0\}$ and

$$y^\top R \, y = (By)^\top (By) = \|w\|^2, \quad y^\top B^\top A B \, y = (By)^\top A(By) = w^\top A \, w \,.$$

Hence

$$\lambda_{\max}(A|_{\mathcal{V}}) = \max_{w \in \mathcal{V}, \, w \neq 0} \frac{w^\top A \, w}{w^\top w}, \qquad \lambda_{\min}(A|_{\mathcal{V}}) = \min_{w \in \mathcal{V}, \, w \neq 0} \frac{w^\top A \, w}{w^\top w} \,.$$

Applying the Courant–Fischer variational characterisation to the ordinary Rayleigh quotient on the proper subspace $\mathcal{V} \subset \mathbb{R}^n$ gives the chain of inequalities $(*)$. If, in addition, $E_{\max}(A) \cap \mathcal{V} = \{0\}$ (respectively $E_{\min}(A) \cap \mathcal{V} = \{0\}$), then the extremal value attained over $\mathcal{V}$ is strictly smaller (respectively strictly larger) than the ambient one. ∎

**Corollary 4 (Euclidean Rayleigh Bounds via Gram-Matrix Extremes)** *Keep the notation and assumptions of Lemma 4 and set*

$$m^{(R)} := \lambda_{\min}(R), \qquad M^{(R)} := \lambda_{\max}(R) \qquad (0 < m \leq M) \,.$$

*Define the* coordinate *Rayleigh quotients*

$$\widehat{\lambda_{\max}}(B^\top AB) := \max_{y \in \mathbb{R}^k, \, y \neq 0} \frac{y^\top B^\top A B \, y}{y^\top y}, \qquad \widehat{\lambda_{\min}}(B^\top AB) := \min_{y \in \mathbb{R}^k, \, y \neq 0} \frac{y^\top B^\top A B \, y}{y^\top y} \,.$$

*Then*

$$m^{(R)} \lambda_{\min}(A) \leq \widehat{\lambda_{\min}}(B^\top AB) \leq \widehat{\lambda_{\max}}(B^\top AB) \leq M^{(R)} \lambda_{\max}(A) \,. \tag{$*$}$$

*Moreover, the right (resp. left) inequality in $(*)$ is strict if and only if $E_{\max}(A) \cap \mathcal{V} = \{0\}$ (resp. $E_{\min}(A) \cap \mathcal{V} = \{0\}$).*

**Proof**  Because $m^{(R)} I \preceq R \preceq M^{(R)} I$, for every $y \neq 0$ one has

$$m^{(R)} y^\top y \leq y^\top R \, y \leq M^{(R)} y^\top y \,.$$

Combining this with the definition of the generalised Rayleigh quotient gives

$$\frac{y^\top B^\top A B \, y}{y^\top y} \leq M^{(R)} \frac{y^\top B^\top A B \, y}{y^\top R \, y}, \qquad \frac{y^\top B^\top A B \, y}{y^\top y} \geq m^{(R)} \frac{y^\top B^\top A B \, y}{y^\top R \, y} \,.$$

Taking the maximum (respectively minimum) over $y \neq 0$ and invoking Lemma 4 yields

$$m^{(R)} \lambda_{\min}(A|_{\mathcal{V}}) \leq \widehat{\lambda_{\min}}(B^\top AB) \leq \widehat{\lambda_{\max}}(B^\top AB) \leq M^{(R)} \lambda_{\max}(A|_{\mathcal{V}}) \,.$$

Applying again the bounds $\lambda_{\min}(A) \leq \lambda_{\min}(A|_{\mathcal{V}})$ and $\lambda_{\max}(A|_{\mathcal{V}}) \leq \lambda_{\max}(A)$ gives the chain $(*)$. Strictness of the outer inequalities propagates from the corresponding strictness in Lemma 4. ∎

### E.3  Uniform Angle Bounds and Genericity

**Lemma 5 (Local Compactness of the Intersection $\mathcal{S} \cap \mathcal{M}_F$ in a bounded neighbourhood)**  *Let a function $f$ be $C^2$. We define the manifold of local minimisers as*

$$\mathcal{S} = \{x \in \mathbb{R}^n : \nabla f(x) = 0,\ f(x) = f_{\mathcal{S}}\},$$

*and the graph of a continuous mapping $\Psi$ as*

$$\mathcal{M}_F = \{(x_1, \Psi(x_1)) : x_1 \in \mathbb{R}^{n_1}\}.$$

*Also, let $\mathcal{N} \subset \mathbb{R}^{n_1+n_2}$ be an closed and bounded set. Write*

$$\mathcal{S}^{\mathrm{loc}} := \mathcal{S} \cap \mathcal{N}, \quad \mathcal{M}_F^{\mathrm{loc}} := \mathcal{M}_F \cap \mathcal{N}.$$

*Then both $\mathcal{S}^{\mathrm{loc}}$ and $\mathcal{M}_F^{\mathrm{loc}}$ are closed subsets of the compact set $\mathcal{N}$, and therefore*

$$\mathcal{S}^{\mathrm{loc}} \cap \mathcal{M}_F^{\mathrm{loc}},$$

*is compact.*

**Proof**  We can write the solution manifold $\mathcal{S}$ as an intersection of two pre-images

$$\mathcal{S} = f^{-1}(\{f_{\mathcal{S}}\}) \cap (\nabla f)^{-1}(\{0\}).$$

Since $f$ is $C^1$, and $\{f_{\mathcal{S}}\} \subset \mathbb{R}$, $\{0\} \subset \mathbb{R}$ are closed sets, it follows that their respective pre-images are closed in $\mathbb{R}^n$. Hence, $\mathcal{S}$ is a closed set.

Similarly, the feasible manifold can be defined as the pre-image of a function $g : \mathbb{R}^{n_1} \times \mathbb{R}^{n_2} \to \mathbb{R}^{n_2}$ such that
$$\mathcal{M}_F = \{(x_1, x_2) : g(x_1, x_2) := x_2 - \Psi(x_1) = 0\} = g^{-1}(\{0\}),$$
is closed in $\mathbb{R}^{n_1+n_2}$.

Finally, since $\mathcal{S}$ and $\mathcal{M}_F$ are closed, their intersection with the neighbourhood $\mathcal{N}$ is a closed subset of a compact set. Any closed subset of a compact set is compact, and thus $\mathcal{S}^{\mathrm{loc}} \cap \mathcal{M}_F^{\mathrm{loc}}$ is compact.  ∎

**Lemma 6 (Uniform Angle Bound for an Isolated Spectral Subspace)**  *Let $\mathcal{N} \subset \mathbb{R}^n$ be a compact set and let $H : \mathcal{N} \to \mathrm{Sym}(n)$ be a continuous map. Let $\mathcal{M}_F \subset \mathbb{R}^n$ be a smooth submanifold such that $K := \mathcal{N} \cap \mathcal{M}_F$ is compact.*

*For each $x \in K$ choose an eigenvalue $\lambda(x) \in \mathrm{spec}(H_x)$ (with multiplicity $k \geq 1$) and assume it is uniformly isolated:*
$$\min_{\mu \in \mathrm{spec}(H_x)\setminus\{\lambda(x)\}} |\mu - \lambda(x)| \ \geq\ \Delta\ >\ 0.$$

*Define the (possibly multi-dimensional) eigenspace*

$$E(x) := \ker\big(H_x - \lambda(x)I\big),$$

*and assume that it satisfies*

$$E(x) \cap T_x\mathcal{M}_F = \{0\}, \qquad \forall\, x \in K.$$

*Then there exists $\theta > 0$ such that for every $x \in K$ and every unit vector $v \in E(x)$ one has*

$$\angle\big(v,\ \mathrm{T}_x\mathcal{M}_F\big) \geq \theta.$$

**Proof**  *Step 1. Continuous subspace field produced by the Riesz projector.* By hypothesis, for every $x \in \mathcal{N} \cap \mathcal{M}_F$ we have

$$\min_{\mu \in \mathrm{spec}(H_x)\setminus\{\lambda(x)\}} |\mu - \lambda(x)| \ \geq\ \Delta\ >\ 0 \quad \forall\, x \in \mathcal{N} \cap \mathcal{M}_F.$$

Hence one can choose a disjoint small closed contour $\gamma(x)$ of radius $\frac{1}{2}\Delta$ around $\lambda(x)$. By construction this contour encloses precisely the $\lambda$-cluster and avoids all other eigenvalues of $H_x$. Then, the Riesz projection onto the $\lambda$–eigenspace,

$$\mathrm{R}(x) = \frac{1}{2\pi i} \oint_\gamma (z - H_x)^{-1}\, dz,$$

is well-defined and depends continuously on $x$ (cf. Kato [80] Ch. II §1.4).

Set

$$E(x) = \operatorname{im} R(x).$$

Thus $x \mapsto E(x)$ is a continuous map $\mathcal{N} \cap \mathcal{M}_F \to \operatorname{Gr}(\mathbb{R}^n)$ into the Grassmannian *i.e.*, it forms a continuous field of subspaces.

*Step 2. Compact unit-sphere field.* Define

$$\mathcal{K} = \left\{ (x, v) : x \in \mathcal{N} \cap \mathcal{M}_F, \, v \in E(x), \, \|v\| = 1 \right\}.$$

Continuity of $x \mapsto E(x)$ implies that the set is closed in the product $\mathcal{N} \cap \mathcal{M}_F \times \mathbb{S}^{n-1}$; because the base $\mathcal{N} \cap \mathcal{M}_F$ is compact (since $\mathcal{M}_F$ is closed - see Lemma 5). Hence, the set $\mathcal{K}$ is compact.

*Step 3. Uniform angle gap.* For any $(x, v) \in \mathcal{K}$, we have $v \notin \mathrm{T}_x \mathcal{M}_F$ by the hypothesis $E(x) \cap \mathrm{T}_x \mathcal{M}_F = \{0\}$. We note that this trivial intersection is generic (Lemma 7). Because orthogonal projection decreases norm unless the vector is in the target space,

$$\varphi(x, v) := \|P_{\mathrm{T}_x \mathcal{M}_F}(v)\| < 1.$$

The function $\varphi : \mathcal{N} \cap \mathcal{M}_F \to [0, 1)$ is continuous; compactness of $\mathcal{K}$ gives a global maximum $\delta = \max_{\mathcal{K}} \varphi < 1$ (by the extreme value theorem). Choosing $\theta = \arccos \delta > 0$ yields $\angle(v, \mathrm{T}_x \mathcal{M}_F) \geq \theta$ for all $(x, v) \in \mathcal{K}$, as claimed. ∎

**Corollary 5 (Uniform Angle for the $\lambda_{\min}$–Eigenspace)** *Let $K := \mathcal{N} \cap \mathcal{S} \cap \mathcal{M}_F \subset \mathbb{R}^n$, where $\mathcal{N}$ is a closed bounded neighbourhood of a local minimiser of $f \in C^2$, $\mathcal{S}$ is the Morse–Bott critical locus of $f$, and $\mathcal{M}_F$ is a smooth submanifold (so $K$ is compact - see Lemma 5). Set*

$$H_x := \nabla^2 f(x)\big|_{N_x \mathcal{S}}, \qquad x \in K,$$

*so that $H_x$ is positive–definite. Suppose*

> *(a) the smallest eigenvalue $\lambda_{\min}(x)$ (mult. $m \geq 1$) of $H_x$ is isolated by a uniform gap*
>
> $$\lambda_{n-m}(x) - \lambda_{\min}(x) \geq \Delta_{\min} > 0,$$

> *(b) its eigenspace $E_{\min}(x) := \ker\big(H_x - \lambda_{\min}(x) I\big)$ satisfies the trivial intersection*
>
> $$E_{\min}(x) \cap \mathrm{T}_x \mathcal{M}_F = \{0\}, \qquad \forall x \in K.$$

*Then Lemma 6 yields a constant $\theta > 0$ such that, for every $x \in K$ and every unit $v \in E_{\min}(x)$,*

$$\angle\big(v, \mathrm{T}_x \mathcal{M}_F\big) \geq \theta.$$

**Corollary 6 (Uniform Angle for the $\sigma_{\max}$–Singular Subspace)** *Let $K := \mathcal{N} \cap \mathcal{M}_F \subset \mathbb{R}^n$, where $\mathcal{N}$ is a closed bounded neighbourhood of a local minimiser of $f \in C^2$ and $\mathcal{M}_F$ is a smooth submanifold (so $K$ is compact - see Lemma 5). For $x \in K$ set*

$$H_x := \nabla^2 f(x), \qquad \sigma_{\max}(x) := \text{largest singular value of } H_x.$$

*Assume*

> *(a) the largest singular value $\sigma_{\max}(x)$ (mult. $p \geq 1$) of $H_x$ is isolated by a uniform gap*
>
> $$\sigma_{\max}(x) - \sigma_{p+1}(x) \geq \Delta_{\max} > 0,$$

> *(b) the eigenspace $E_{\max}(x) := \ker\big(H_x^2 - \sigma_{\max}^2(x) I\big)$ satisfies the trivial intersection*
>
> $$E_{\max}(x) \cap \mathrm{T}_x \mathcal{M}_F = \{0\}, \qquad \forall x \in K.$$

*Then, by Lemma 6 (with the eigenvalue cluster $\{\sigma_{\max}^2(x)\}$ of $H_x^2$), there exists $\theta > 0$ such that for every $x \in K$ and every unit $v \in E_{\max}(x)$,*

$$\angle\big(v, \mathrm{T}_x \mathcal{M}_F\big) \geq \theta.$$

**Lemma 7 (Genericity of Trivially-Intersecting Subspaces)** *Let $0 < k, l < N$ with $k + l < N$. Write $\mathrm{Gr}(k, N)$ (resp. $\mathrm{Gr}(l, N)$) for the real Grassmannian of k–planes (resp. l–planes) in $\mathbb{R}^N$ and set*

$$\Sigma = \{(\mathcal{U}, \mathcal{V}) \in \mathrm{Gr}(k, N) \times \mathrm{Gr}(l, N) \mid \mathcal{U} \cap \mathcal{V} \neq \{0\}\}.$$

*Then $\Sigma$ is a real-algebraic subset of $\mathrm{Gr}(k, N) \times \mathrm{Gr}(l, N)$ of codimension*

$$\mathrm{codim}\, \Sigma = N - (k + l) + 1 \ \geq 1.$$

*Consequently $\Sigma$ has empty interior and Lebesgue measure $0$; in particular a generic pair of planes satisfies $\mathcal{U} \cap \mathcal{V} = \{0\}$.*

**Proof** *Step 1. Rank-drop criterion.* Choose full-rank matrices $U \in \mathbb{R}^{N \times k}$, $V \in \mathbb{R}^{N \times l}$ whose column spaces are $\mathcal{U}$ and $\mathcal{V}$, and form the $N \times (k + l)$ matrix $M = [U\ V]$. We have

$$\mathcal{U} \cap \mathcal{V} \neq \{0\} \quad \Longleftrightarrow \quad \mathrm{rank}\, M \leq k + l - 1.$$

Thus the "bad" locus $\Sigma$ is the image in $\mathrm{Gr}(k, N) \times \mathrm{Gr}(l, N)$ of the determinantal variety

$$D := \{M \in \mathbb{R}^{N \times (k+l)} \mid \mathrm{rank}\, M \leq k + l - 1\},$$

defined by the vanishing of all $(k + l) \times (k + l)$ minors of $M$.

*Step 2. Codimension in matrix space.* For general integers $N, m, r$ with $r < m \leq N$,

$$\dim\{N \times m \text{ matrices of rank } \leq r\} = (N + m)r - r^2,$$

hence

$$\mathrm{codim}_{\mathbb{R}^{N \times m}}\{\mathrm{rank} \leq r\} \ = \ Nm - (N + m)r + r^2 = (N - r)(m - r).$$

Taking $m = k + l$ and $r = k + l - 1$ gives

$$\mathrm{codim}_{\mathbb{R}^{N \times (k+l)}} D = (N - (k + l) + 1).$$

Because $k + l < N$ by hypothesis, this number is $\geq 1$.

*Step 3. Passage to the Grassmannians.* The product of Stiefel manifolds $\mathrm{St}(k, N) \times \mathrm{St}(l, N) \subset \mathbb{R}^{N \times (k+l)}$, is an open subset of the full-rank matrices, so intersecting $D$ with it cannot decrease codimension. Next, the quotient map

$$\mathrm{St}(k, N) \times \mathrm{St}(l, N) \longrightarrow \mathrm{Gr}(k, N) \times \mathrm{Gr}(l, N), \quad (Q_1, Q_2) \mapsto (\mathrm{col}(Q_1), \mathrm{col}(Q_2)),$$

is a smooth submersion with compact fibres $O(k) \times O(l)$, which preserves codimension. Consequently

$$\mathrm{codim}_{\mathrm{Gr}(k, N) \times \mathrm{Gr}(l, N)} \Sigma = N - (k + l) + 1.$$

*Step 4. Genericity.* Because $\Sigma$ is a proper real-algebraic subset of positive codimension, it has Lebesgue measure $0$ and empty interior. Its complement is therefore dense (indeed Zariski open) and of full measure, so a generic pair $(\mathcal{U}, \mathcal{V})$ satisfies $\mathcal{U} \cap \mathcal{V} = \{0\}$. ∎

### E.4  Genericity of Single Eigenvalues

**Lemma 8 (Codimension of the Repeated Eigenvalue Locus [81, 80])** *Let*

$$S(n) = \{A \in \mathbb{R}^{n \times n} : A = A^\top\},$$

*be the space of real symmetric $n \times n$ matrices, and let,*

$$\Sigma = \{A \in S(n) : A \text{ has a repeated eigenvalue}\}.$$

*Then $\Sigma$ is an algebraic subset of $S(n)$ (namely, the zero set of the discriminant of the characteristic polynomial).*

*At a matrix $A_0 \in \Sigma$ where an eigenvalue $\lambda_0$ has multiplicity exactly $2$ (with the other eigenvalues distinct and different from $\lambda_0$), the condition that a nearby matrix has a double eigenvalue imposes two independent (local) constraints on its parameters.*

*Consequently, the codimension of $\Sigma$ in $S(n)$ is at least $2$.*

**Remark 5 (Generic Simplicity of the Eigenvalues of Symmetric Matrices)** *Lemma 8 shows that the set of symmetric matrices with repeated eigenvalues is a real-analytic subset of $\mathcal{S}_n$, which can be decomposed into a finite union of smooth submanifolds, each of codimension at least two. Since smooth submanifolds of codimension $\geq 2$ have Lebesgue measure zero (by applying Sard's theorem [82]), it follows that this set has measure zero. Therefore, the complement—the set of symmetric matrices with simple (distinct) eigenvalues—has full measure. In other words, simple eigenvalues are generic among symmetric matrices.*

## F  Transversality and Clean Intersection of Smooth Manifolds

To study the geometry of constrained critical sets, we begin by recalling two foundational concepts that govern how smooth manifolds intersect: transversality [83, Chapter 3] and clean intersection [84, Appendix C.3]. These notions describe the local structure and regularity of the intersection and play a central role in establishing genericity results for solution sets in optimisation and variational problems. To build geometric intuition, we illustrate examples of transverse and clean (but non-transverse) intersections in Figure 5.

Let
$$\mathcal{M}, \ \mathcal{N} \ \subset \ \mathcal{Z} \,,$$
be embedded submanifolds of a smooth manifold $\mathcal{Z}$, with
$$\dim \mathcal{M} = r, \quad \dim \mathcal{N} = k, \quad \dim \mathcal{Z} = l \,.$$

The classical condition of transversality ensures that two manifolds intersect in general position, meaning their tangent spaces at each point of intersection span the ambient space. This condition is central in differential topology and guarantees that intersections behave stably under perturbation. However, in many geometric and optimisation contexts, transversality is unnecessarily strong. For example, if the sum of the dimensions satisfies $r + k < l$, then transversality cannot hold unless the intersection is empty. Consequently, generic perturbations that enforce transversality may eliminate meaningful intersections entirely.

This behaviour is undesirable in applications where the intersection encodes feasible or optimal solutions—properties we wish to preserve. To accommodate more flexible and structured intersections, the notion of clean intersection provides a weaker but still geometrically meaningful alternative. Clean intersection allows the tangent spaces to align nontrivially, as long as the intersection remains a smooth submanifold with compatible tangent structure. This relaxation offers a more flexible framework that preserves non-empty intersections under generic, local perturbations, which is exactly the setting we consider in our problem.

We now formalise both notions:

**Definition 4 (Transversality)**  *We say that $\mathcal{M}$ and $\mathcal{N}$ intersect transversally, denoted by $\mathcal{M} \pitchfork \mathcal{N}$, if for every point $x \in \mathcal{M} \cap \mathcal{N}$ the tangent spaces satisfy*
$$\mathrm{T}_x \mathcal{M} + \mathrm{T}_x \mathcal{N} = \mathrm{T}_x \mathcal{Z} \,,$$
*or, equivalently, the normal spaces satisfy*
$$\mathrm{N}_x \mathcal{M} \cap \mathrm{N}_x \mathcal{N} = \{0\} \,.$$

**Definition 5 (Clean Intersection)**  *We say that $\mathcal{M}$ and $\mathcal{N}$ intersect cleanly, denoted by $\mathcal{M} \pitchfork \mathcal{N}$, if for every point $x \in \mathcal{M} \cap \mathcal{N}$, we have that the dimension $d := \dim(\mathrm{T}_x \mathcal{M} \cap \mathrm{T}_x \mathcal{N})$ is constant and*
$$\mathrm{T}_x (\mathcal{M} \cap \mathcal{N}) = \mathrm{T}_x \mathcal{M} \cap \mathrm{T}_x \mathcal{N} \,.$$

**Remark 6**  *Transversality implies a clean intersection between manifolds but the converse is not always true.*

To rigorously analyse generic properties of smooth manifolds and function spaces, we work within the topological framework of residual sets. This notion allows us to make precise what it means for a property to hold "generically"—that is, to be true for a large and stable class of objects. In particular, we will be concerned with residual subsets in spaces of smooth functions or immersions, equipped with the Whitney $C^r$ topology.

**Definition 6 (Residual set & generic property)** *Let $\mathcal{X}$ be a topological space. A subset*

$$A \subset \mathcal{X},$$

*is called* residual *if it contains a countable intersection of dense open sets in $\mathcal{X}$. If $\mathcal{X}$ is a Baire space (e.g., any Banach or Fréchet space, with its usual topology), then every residual set is itself dense in $\mathcal{X}$.*

*A property $\mathscr{P}$ of points in $\mathcal{X}$ is said to hold* generically *if the set*

$$\{x \in \mathcal{X} : x \text{ satisfies } \mathscr{P}\},$$

*is residual in $\mathcal{X}$.*

Within this framework, differential topology offers a foundational result: transversality is a generic condition, as we see in the following Lemma. That is, for a large class of smooth mappings, transversality to a fixed submanifold holds generically under perturbation.

**Lemma 9 (Generic Transversality of Submanifolds)** *Let $\mathcal{M}, \mathcal{N} \subset \mathcal{Z}$ be submanifolds of a smooth manifold $\mathcal{Z}$, and let $\mathscr{F}$ be the space of smooth immersions of one of the submanifolds into $\mathcal{Z}$ (e.g., $\mathcal{M}$). Then, the set of maps $\phi \in \mathscr{F}$ such that $\phi(\mathcal{M}) \pitchfork \mathcal{N}$ is residual in $\mathscr{F}$ with the Whitney $C^r$ topology for $r \geq 1$. In particular transverse intersection is generic under smooth perturbation.*

**Proof** This result follows directly from standard transversality theory, specifically the Parametric Transversality Theorem or Thom's Transversality Theorem. See, for example, [83, Chapter 3, Theorems 2.1 and 2.9], where it is shown that the set of maps transverse to a fixed submanifold is residual in Whitney $C^r$ topology for any $r \geq 1$. ∎

We now apply these concepts to the setting of critical points. Let $f \in C^2(\mathbb{R}^n)$ be a smooth function satisfying the Morse–Bott condition, and consider the set of critical points at a fixed value as defined below. We are particularly interested in the intersection of this critical locus with a fixed feasible manifold $\mathcal{M}_F \subset \mathbb{R}^n$. While transverse intersection may fail due to dimension constraints, clean intersection remains a viable and meaningful condition—and, crucially, it holds generically within the space of $C^2$ functions.

**Definition 7 (Critical-locus map)** *Fix $c \in \mathbb{R}$ and a $C^2$ function $f : \mathbb{R}^n \to \mathbb{R}$ in which the Morse–Bott property holds. Define*

$$\mathscr{C}(x) = \big(f(x) - c, \nabla f(x)\big) : \mathbb{R}^n \to \mathbb{R} \times \mathbb{R}^n.$$

*Its zero-set $\mathcal{S} = \mathscr{C}^{-1}(0,0)$ is the Morse–Bott critical locus.*

**Lemma 10 (Equivalence of Clean Intersection and Local Transversality [85, Lemma 2.6])** *Let $\mathcal{M}, \mathcal{N} \subset \mathcal{Z}$ be submanifolds of a smooth manifold $\mathcal{Z}$, and suppose that the dimension $\dim(\mathrm{T}_x\mathcal{M} \cap \mathrm{T}_x\mathcal{N})$ is locally constant for all $x \in \mathcal{M} \cap \mathcal{N}$. Then, the following are equivalent:*

1. *$\mathcal{M} \pitchfork \mathcal{N}$;*
2. *For each $x \in \mathcal{M} \cap \mathcal{N}$, there exists a submanifold $\mathcal{N}' \subset \mathcal{Z}$ containing $\mathcal{N}$ near $x$, with $\mathrm{codim}(\mathcal{N}') = \dim \mathcal{M} - \dim(\mathrm{T}_x\mathcal{M} \cap \mathrm{T}_x\mathcal{N})$ such that*

$$\mathcal{M} \pitchfork \mathcal{N}' \quad and \quad \mathcal{M} \cap \mathcal{N} = \mathcal{M} \cap \mathcal{N}' \quad \text{locally near } x.$$

*In particular, clean intersection between $\mathcal{M}$ and $\mathcal{N}$ near $x$ is equivalent to transversality of $\mathcal{M}$ with an auxiliary submanifold $\mathcal{N}'$ that locally agrees with $\mathcal{N}$ on the intersection.*

**Lemma 11 (Generic Clean Intersection of the Critical Locus)** *Let $\mathcal{S} = \mathscr{C}^{-1}(0,0)$ be the Morse–Bott critical locus of a $C^2$ function $f : \mathbb{R}^n \to \mathbb{R}$ at level $c \in \mathbb{R}$ as in Definition 7, and let $\mathcal{M}_F \subset \mathbb{R}^n$ be a fixed embedded submanifold. Then, clean intersection between $\mathcal{S}$ and $\mathcal{M}_F$ is a generic property in the space of $C^2$ functions under the Whitney topology.*

**Proof** By Lemma 10, clean intersection between $\mathcal{S}$ and $\mathcal{M}_F$ near a point $x \in \mathcal{S} \cap \mathcal{M}_F$ is equivalent to the existence of a submanifold $\mathcal{N} \subset \mathbb{R}^n$ containing $\mathcal{M}_F$ near $x$ such that

$$\mathcal{S} \cap \mathcal{M}_F = \mathcal{S} \cap \mathcal{N} \quad \text{locally}, \quad \text{and} \quad \mathcal{S} \pitchfork \mathcal{N}.$$

That is, clean intersection is locally equivalent to transversality between $\mathcal{S}$ and an auxiliary submanifold $\mathcal{N}$.

Since $\mathcal{S}$ is defined as the zero set of the critical-locus map $\mathscr{C}(x) = (f(x) - c, \nabla f(x))$, it varies smoothly with $f$ in the Whitney $C^2$ topology. By Lemma 9, transversality $\mathcal{S} \pitchfork \mathcal{N}$ is a generic condition among $C^2$ functions. Therefore, clean intersection $\mathcal{S} \pitchfork \mathcal{M}_F$ holds generically in the space of $C^2$ functions. ∎

**Remark 7 (Density and Small Perturbations)** *Since every residual set in a Baire space is dense, the generic set*

$$\{ f \in C^2(\mathbb{R}^n) \; : \; \mathrm{Crit}_c(f) \pitchfork \mathcal{M}_F \},$$

*is not only large in the topological sense but also satisfies: for any given $f$ and any $\epsilon > 0$, there exists a $g \in C^2(\mathbb{R}^n)$ with $\|g - f\|_{C^2} < \epsilon$ such that $\mathrm{Crit}_c(g) \pitchfork \mathcal{M}_F$.*

*In other words, any non-generic $f$ can be made generically clean by an arbitrarily small $C^2$ perturbation, without making the intersection empty.*

# G Experimental Validation of Results

To validate our theoretical findings, we present three illustrative examples. These examples demonstrate the application and performance of our proposed reduction mapping method in different optimisation scenarios, from a simple 2D quadratic case to a more complex high-dimensional nonlinear problem.

## G.1 Example 1: 2D Quadratic Problem

We begin with a simple two-dimensional quadratic function to visualise the trajectory of the optimisation process. It follows the first example of the Appendix A, where the objective function is given by,

$$G(x, y) = x^2 + 10(y - x)^2.$$

The reduction mapping, which constrains the optimisation to the manifold where $y = x$, is defined as,

$$\Phi(x) = \begin{pmatrix} x \\ x \end{pmatrix}.$$

The reduced function is $F(x) = x^2$, with Hessian $\nabla^2 F(x) = 2$. In this case, the pullback metric tensor $R = D\Phi^\top D\Phi = 2$ is identical to the Hessian. Consequently, our geometrically preconditioned gradient descent method recovers the full Newton method. As expected for a quadratic function, this achieves convergence in a single step. Figure 6 illustrates the optimisation paths on the contour plot of the function. We can observe that the reduced method directly follows the valley of the function, leading to faster convergence. The learning rates for each method are set based on the smoothness constant of the function, *i.e.*, $\eta = 1/\beta$.

## G.2 Example 2: High-Dimensional Quadratic Problem

Next, we consider a high-dimensional quadratic problem ($n = 40$). The objective function is,

$$G(x, y) = \frac{1}{2}\|x\|^2 + \frac{1}{2}\|y\|^2 + \frac{\lambda}{2}\|y - Kx\|^2,$$

where $x \in \mathbb{R}^n$, $y \in \mathbb{R}^n$, and $K$ is a linear operator. We also set the parameter $\lambda$ equal to ten and keep it fixed during training. The reduction mapping is given by the manifold where $y = Kx$:

$$\Phi(x) = \begin{pmatrix} x \\ Kx \end{pmatrix}.$$

The gradient and Hessian of the original function are,

$$\nabla G(x, y) = \begin{bmatrix} x + \lambda K^\top(Kx - y) \\ y + \lambda(y - Kx) \end{bmatrix}, \quad \text{and} \quad \nabla^2 G(x, y) = \begin{bmatrix} I + \lambda K^\top K & -\lambda K^\top \\ -\lambda K & (1 + \lambda)I \end{bmatrix}.$$

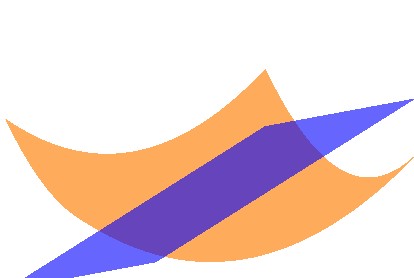

Figure 5a: Transverse Intersection

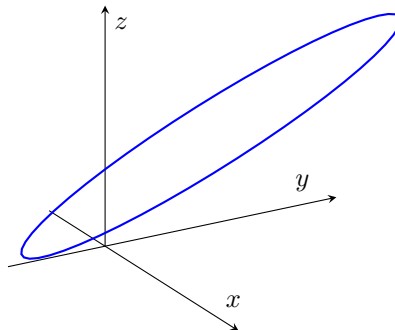

Figure 5b: Corresponding Intersection Set

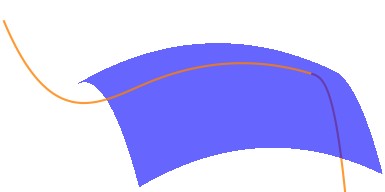

Figure 5c: Clean & Non-Transverse

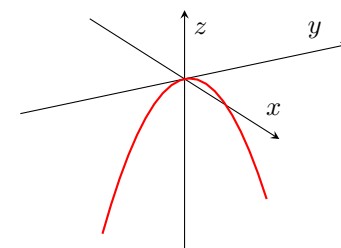

Figure 5d: Corresponding Intersection Set

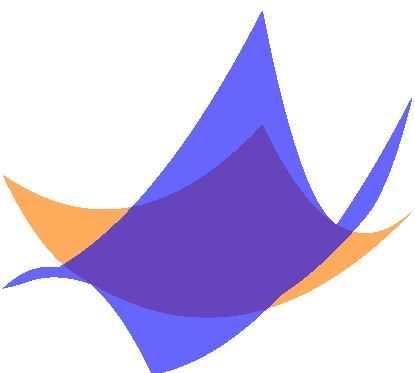

Figure 5e: Non-Clean & Non-Transverse

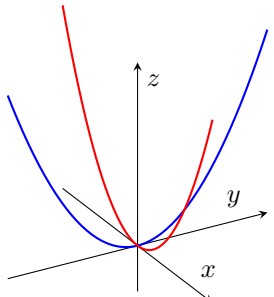

Figure 5f: Corresponding Intersection Set

Figure 5: Visualisations of three types of surface intersections in $\mathbb{R}^3$. Left column shows intersecting surfaces, right column shows corresponding intersection curves. **Top (Transverse)**: The paraboloid $z = x^2 + y^2$ intersects the plane $z = 1 + x + y$ transversely, with distinct tangent planes at every point of intersection. **Middle (Clean, Non-Transverse)**: The surface $z = -0.02x^2 - 0.02y^2$ is intersected by the curve $\gamma(t) = (0, \ t, \ -0.02t^2 + \varphi(t))$, where $\varphi(t) = 0$ for $|t| \leq 1$, $\varphi(t) = 0.04(-t-1)^3$ for $t < -1$, and $\varphi(t) = -2(t-1)^3$ for $t > 1$. The curve lies exactly on the surface for $t \in [-1, 1]$, so the intersection is a smooth 1D submanifold and hence clean. However, outside this interval, the curve deviates from the surface, and the tangent vectors do not span $\mathbb{R}^3$, so the intersection is not transverse. **Bottom (Non-Clean, Non-Transverse)**: The surfaces $z = x^2 + y^2$ and $z = x^2 + y^2 + x^2y$ intersect along $\{x = 0\} \cup \{y = 0\}$, forming two parabolic curves that meet at the origin. At the origin, there is a singularity, so the intersection is not a smooth submanifold, hence neither clean nor transverse.

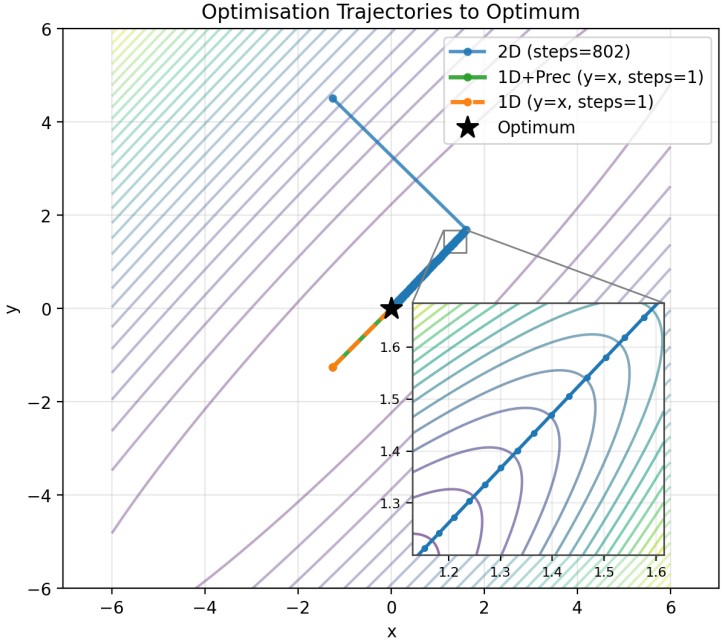

Figure 6: Optimisation trajectories for the 2D quadratic problem.

The reduced function $F(x) = G(\Phi(x))$ simplifies to,

$$F(x) = \frac{1}{2}\|x\|^2 + \frac{1}{2}\|Kx\|^2 \,.$$

The gradient and Hessian of the reduced function are,

$$\nabla F(x) = x + K^\top K x, \quad \text{and} \quad \nabla^2 F(x) = I + K^\top K \,.$$

For this linear least-squares problem, the Hessian of the reduced function is identical to the pullback metric $R = D\Phi^\top D\Phi = I + K^\top K$. As in the first example, our geometrically preconditioned gradient descent is equivalent to the full Newton method applied to the reduced function, which converges in a single iteration. The experimental results for this case are summarised in Figures 7, 8, and, 9. The learning rates for each method are set based on Armijo's backtracking line search.

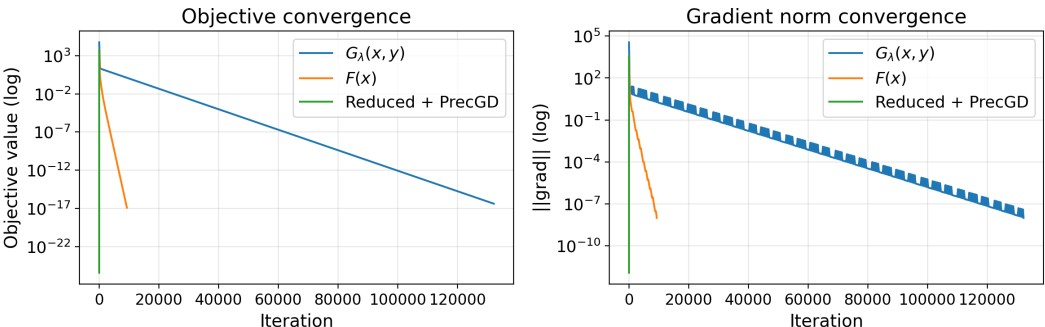

Figure 7: Convergence rate for the high-dimensional quadratic problem.

## G.3 Example 3: High-Dimensional Nonlinear Problem

Finally, we address a high-dimensional nonlinear problem ($n = 40$), with the same settings as before. The objective function is,

$$G(x, y) = \frac{1}{2}\|x\|^2 + \frac{1}{2}\|y\|^2 + \frac{\lambda}{2}\|y - \alpha \tanh(Kx)\|^2 \,.$$

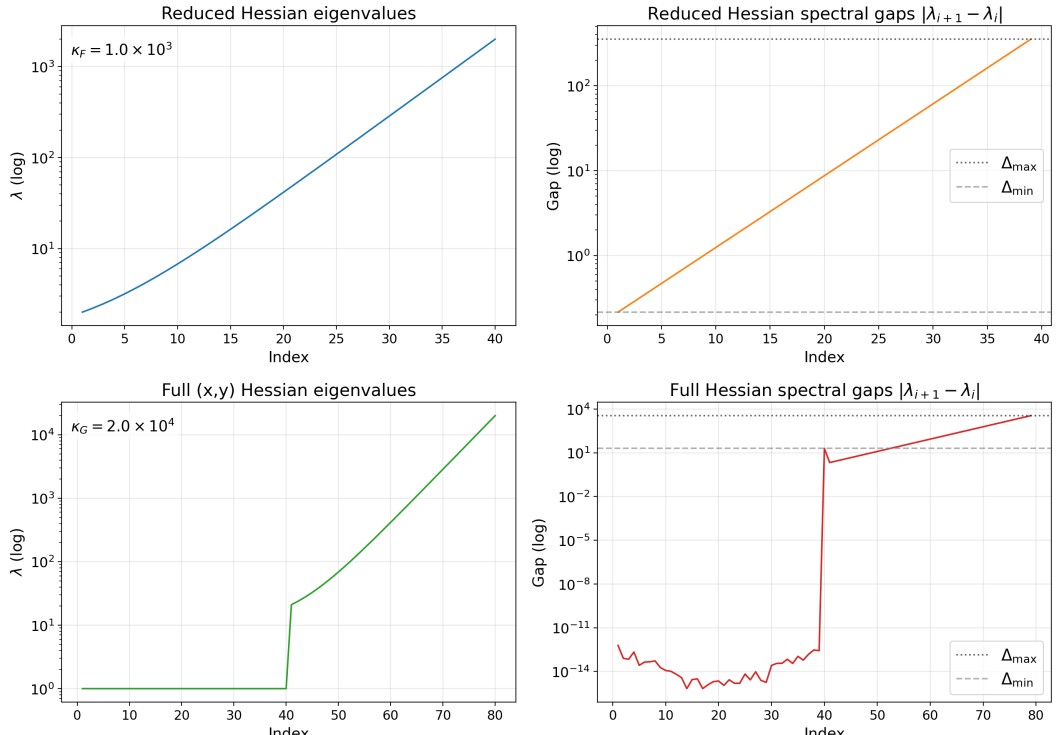

Figure 8: Hessian eigenspectrum for the high-dimensional quadratic problem.

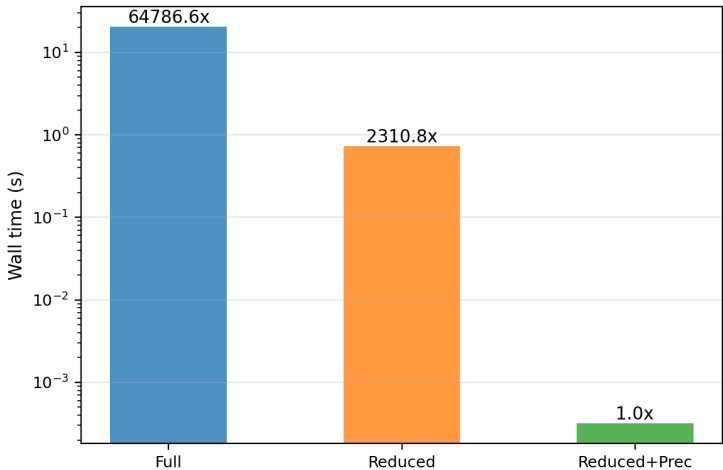

Figure 9: Wall-clock time for the high-dimensional quadratic problem. This figure compares the wall-clock time required to reach convergence.

First, let us define intermediate variables,

$$u = Kx, \quad t = \tanh(u), \quad s = \operatorname{sech}^2(u) = 1 - t^2, \quad r = y - \alpha t.$$

The gradient is,

$$\nabla G(x,y) = \begin{bmatrix} x - \lambda \alpha K^\top (\operatorname{Diag}(s)r) \\ y + \lambda r \end{bmatrix}.$$

The Hessian is,

$$\nabla^2 G(x,y) = \begin{bmatrix} \nabla_{xx}^2 G(x,y) & \nabla_{xy}^2 G(x,y) \\ \nabla_{yx}^2 G(x,y) & \nabla_{yy}^2 G(x,y) \end{bmatrix},$$

where,

$$\nabla^2_{xx}G(x,y) = I + \lambda\alpha^2 K^\top \text{Diag}(s)^2 K + \lambda\alpha K^\top \text{Diag}(g)K$$
$$\nabla^2_{xy}G(x,y) = -\lambda\alpha K^\top \text{Diag}(s)$$
$$\nabla^2_{yx}G(x,y) = -\lambda\alpha \text{Diag}(s)K$$
$$\nabla^2_{yy}G(x,y) = (1+\lambda)I\,.$$

In the $\nabla^2_{xx}G$ block, $g$ is a vector defined as $g = 2r \odot s \odot t$.

The reduction mapping is defined by the nonlinear manifold $y = \alpha\tanh(Kx)$,

$$\Phi(x) = \begin{pmatrix} x \\ \alpha\tanh(Kx) \end{pmatrix}\,.$$

The reduced function is,

$$F(x) = \frac{1}{2}\|x\|^2 + \frac{1}{2}\|\alpha\tanh(Kx)\|^2\,.$$

This problem structure, minimising $\frac{1}{2}\|\Phi(x)\|^2$, is a nonlinear least-squares problem. The gradient and Hessian of the reduced function are given below,

$$\nabla F(x) = x + \alpha^2 K^\top \text{Diag}(s)t \quad\text{and}\quad \nabla^2 F(x) = I + \alpha^2 K^\top \text{Diag}(g)K\,,$$

where $g = s \odot s - 2t \odot t \odot s$. The pullback metric $R = D\Phi^\top D\Phi$ is given by,

$$R = I + \alpha^2 K^\top \text{Diag}(s^2)K\,.$$

In this setting, our geometrically preconditioned gradient descent, which uses $R$ as the preconditioner, is precisely equivalent to the Gauss-Newton method applied to the reduced function. This is a special case as the the function is a nonlinear least-squares problem. The experimental results for the nonlinear case are presented in Figures 10, 11, and, 12. The learning rates for each method are set based on Armijo's backtracking line search.

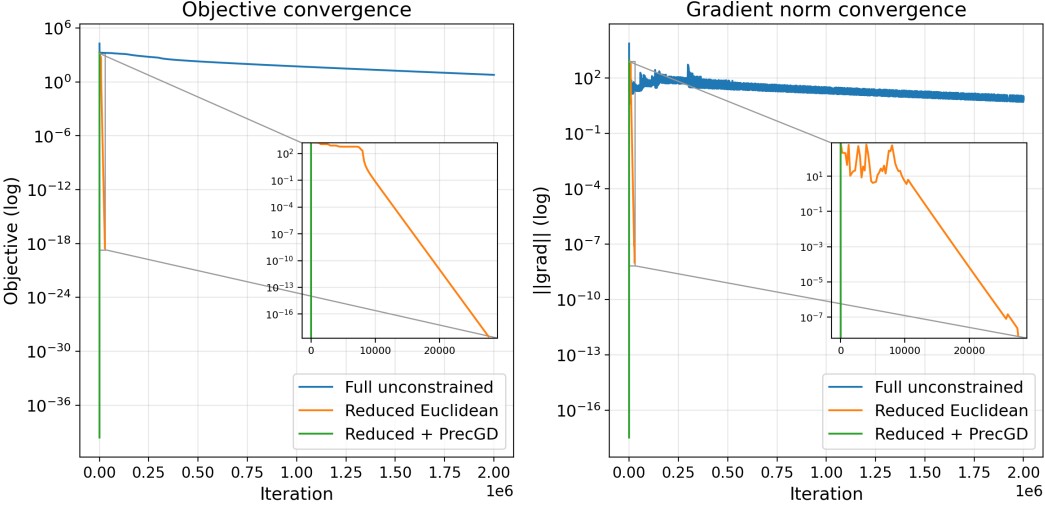

Figure 10: Convergence rate for the high-dimensional nonlinear problem.

These examples demonstrate the flexibility of our proposed framework. While it is general enough to apply to any objective function $G$ and mapping $\Phi$, these experiments show that for common problem classes, it naturally recovers well-known, powerful optimisation algorithms such as the Newton and Gauss-Newton methods.

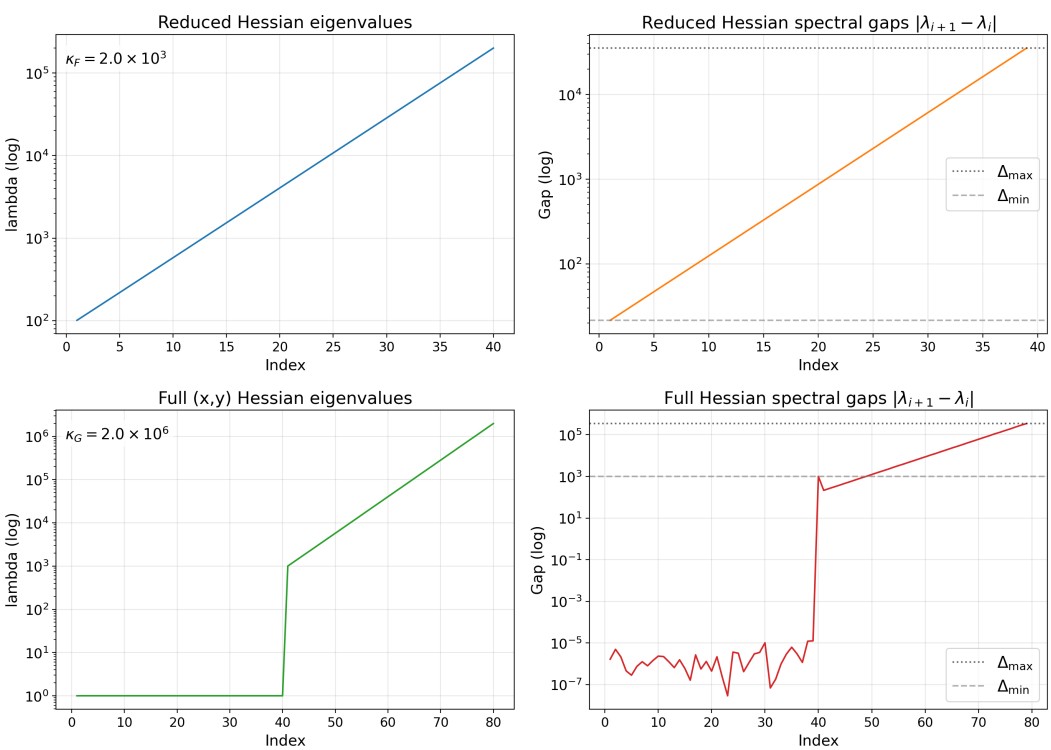

Figure 11: Hessian eigenspectrum for the high-dimensional nonlinear problem.

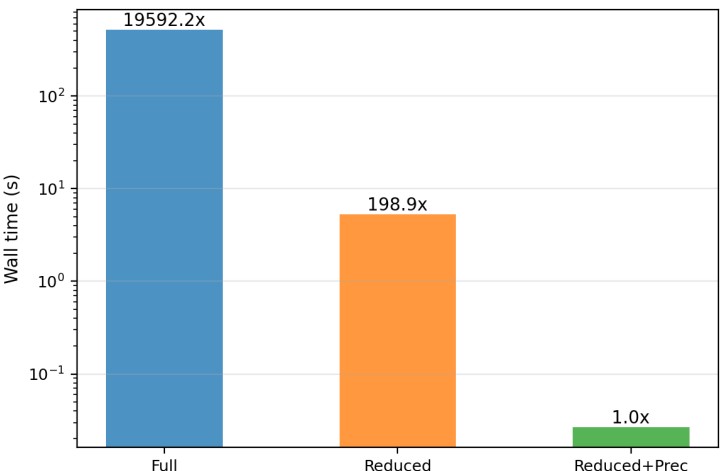

Figure 12: Wall-clock time for the high-dimensional nonlinear problem, comparing the time to convergence. Note that the Reduced+PrecGD wall-clock time also includes the conjugate gradient steps needed to solve for the preconditioning at each iteration.

