# OpenReview forum: "Sharper Convergence Rates for Nonconvex Optimisation via Reduction Mappings"
_NeurIPS.cc/2025/Conference — NeurIPS 2025 spotlight_

### Official Review · Reviewer_1oNF · 2025-06-30

**Clarity:** 4
**Significance:** 2
**Originality:** 3
**Rating:** 5
**Confidence:** 3

**Summary:**

This paper is about using reduction mappings to exploit geometric structure of the solution set to accelerate convergence. It characterizes when reduction mappings lead to provable gains in optimization efficiency, i.e. optimizing on a manifold with a landscape that makes optimization easier for first-order methods. They show that under affine or nonlinear reduction mappings, the smoothness (Lipschitz) constant of the reduced function $F$ is strictly smaller than that of the original function.$f$. Under nonlinear reduction mappings, they also show that the sharpness constant is improved. Thus, the better condition number $\kappa$ of $F$ over $f$ and faster convergence through the use of methods such as gradient descent where the rate depends on $\kappa$.

**Questions:**

- Neural collapse is observed in stochastic settings. Could faster convergence via reduction mappings also be observed empirically?
 - Preconditioning is recommended to take advantage of the structure from reduction mapping, but in practice, exact applications of preconditioning is difficult. Is there an approximate form of the pullback metric $R(x_1)$ that is simpler, e.g. diagonal?

**Ethical Concerns:**

["NO or VERY MINOR ethics concerns only"]

**Final Justification:**

I appreciate the additional experiments and the authors response to my concerns. I maintain my rating at Accept.

**Limitations:**

Yes

**Quality:**

3

**Strengths And Weaknesses:**

Strengths
- Paper is very clearly written. Their framework ties together preconditioned methods, normalization methods, bilevel optimization and other domains via reduction mappings.
 - Authors give provable convergence improvements of nonlinear reduction mappings by showing that both the smoothness and sharpness constants are improved in the reduced

Weaknesses
 - Could be difficult to understand how to use this from a practical standpoint. While this is a theory paper, pseudo-code showing how the algorithm could be implemented in practice and experimental results (even with synthetic data) could make this easier to grasp for practioners.
 - Some limitations were acknowledged by the authors: local analysis only, the assumption of exact evaluation of $\Psi(x_1)$, the unknown (problem dependent) cost of evaluating $\Psi(x_1)$, analysis limited to deterministic setting

---

> ### Author Rebuttal · Authors · 2025-07-30
>
> We thank the reviewer for their thoughtful and constructive feedback. We address each point below.
>
> **Weaknesses**
>
> **1)** We agree that practical clarity is important, even in a theory-focused paper. In response, we have included several synthetic experiments that demonstrate the practical benefits of our approach (see the response to reviewer 1 (6kyE) for details). These include:
> - a 2D toy example (Example 1, Appendix A),
> - a high-dimensional linear problem with ill-conditioned structure, and
> - a high-dimensional nonlinear experiment using a $\tanh$-based mapping.
>
>
> These experiments empirically validate the iteration complexity improvements predicted by our framework.
> Additionally, we highlight that our framework connects with real-world settings such as neural collapse and ETF parametrisation, as explored in prior work [46]. We plan to apply our framework and cover more complex settings and real data applications in future work.
>
> **2)** Indeed, the limitations mentioned—local analysis, assumption of exact mappings, deterministic setting—are important and we fully agree that they offer meaningful avenues for future research. Our current focus was to establish a rigorous foundation, but we view extending the theory to stochastic and approximate regimes as a natural and promising next step.
>
> **Questions:**
>
> **1)** Yes, faster convergence of neural collapse due to reduction mappings has already been observed in stochastic settings in prior work [46]. In particular, the empirical results in that paper show significant gains under SGD when using ETF-reparametrisation mappings. Extending our theoretical framework to analyse convergence in such stochastic regimes is a key future direction.
>
> **2)** We agree that exact application of the pullback metric can be computationally expensive in general. However, in many structured applications, the metric admits efficient forms:
> Linear or affine mappings yield a constant $R$ that is easy to compute and invert.
> Structured nonlinear mappings (e.g. arising from normalisation layers or architectural constraints) often produce diagonal, block-diagonal, or low-rank $R$, enabling efficient approximations.
> For more general nonlinear cases, approximate preconditioning techniques (e.g. diagonal approximation, conjugate gradients, or low-rank updates) can be used to trade off computational cost with fidelity to the pullback geometry.
> These situations are common in practice, and we believe they offer a rich space for practical extensions of our theory, which we intend to explore further in future work.

---

> > ### Comment · Reviewer_1oNF · 2025-08-07
> >
> > I thank the authors for their response and appreciate the inclusion of experimental results. Nevertheless, the additional experimental results are preliminary and would benefit from more rigorous investigation (e.g. inclusion of wall clock time, training convergence plots, etc.). Thus I maintain my score.

---

### Official Review · Reviewer_rmzJ · 2025-07-02

**Clarity:** 3
**Significance:** 3
**Originality:** 4
**Rating:** 5
**Confidence:** 4

**Summary:**

This paper proposes a new framework for analyzing the contribution that the local geometry of minimizers has on the optimization process. The framework is based on reduction mappings that reparameterize the problem in terms of the solution manifold, which reduces the dimensionality of the problem. When the optimization process incorporates a well chosen reduction mapping, the authors prove that the local smoothness and sharpness constants of the objective improve, leading to faster local convergence rates. This framework unifies a number of approaches, such as batch normalization and equiangular tight frame projections in deep neural networks, that utilize reduction mappings and gives a theoretical explanation for the observed benefits of these approaches.

**Questions:**

Are the results tight in the sense of matching previously known bounds? It is mentioned in the appendix that prior work has shown that in the case of batch normalization, the smoothness and sharpness constants are altered, just as in this work. In this case, does the proposed framework generate (roughly) the same improvement in these constants, or does the generality of the framework reduce the tightness of the theoretical results?

The curvature and spectral gap conditions are sufficient to guarantee the improved smoothness and sharpness constants. Is it known if these conditions are also necessary, or are there other conditions that could be satisfied for which a similar improvement is possible?

**Ethical Concerns:**

["NO or VERY MINOR ethics concerns only"]

**Final Justification:**

The authors responded to all of my questions in a satisfactory manner and I maintain my original recommendation of accept.

**Limitations:**

Yes.

**Paper Formatting Concerns:**

No.

**Quality:**

3

**Strengths And Weaknesses:**

Strengths: This paper proposes a novel viewpoint to unify many commonly used techniques in nonconvex optimization that demonstrates the potential theoretical benefit of that incorporating a smartly chosen reduction mapping can have on the local convergence properties of first order methods. The proposed framework is novel and has a high degree of flexibility, allowing it to cover many interesting cases such as batch normalization, ETF reparameterizations, and bilevel optimization. Sufficient conditions are given under which a well chosen parameterization will yield improved local convergence properties for first order methods.

Weaknesses: The paper mentions many potential applications of this framework, however, the authors do not apply their framework to any problem beyond some simple, 2-d examples given in the appendix. While the reduction mapping approach is interesting, it is hard to validate how effective it is at improving the understanding of the methods used without applying it to an example application, such as batch normalization.

Quality: The paper is technically sound and provides an interesting viewpoint on how reduction mappings can have a beneficial impact on the local geometry of optimization problems. The framework is quite general and includes many interesting potential applications. However, no concrete applications of the framework is provided in the paper, which makes it hard to validate how tight the bounds generated by this approach are when compared with a more direct analysis on a case by case basis.

Clarity: On a whole, the paper is well written, however, some of the best motivation and explanation of reduction mappings is present in the appendix (section A). It would be ideal if some of this motivation was worked into the main body of the paper, to improve the introduction into the framework, which quickly becomes technical in the main paper.

Significance: The framework proposed is quite general and appears to apply to many interesting scenarios in nonconvex optimization. It seems that it could have a significant impact on how to view many strategies incorporated in modern optimization methods, though this is not entirely clear, as no concrete examples are evaluated using the proposed framework.

Novelty: To my knowledge, this is the first work to use reduction mappings to unify a variety of techniques currently in use in nonconvex optimization. It provides a unique viewpoint, that could be used to not only better understand methods that are currently in use, but to design new techniques with improved performance.

---

> ### Author Rebuttal · Authors · 2025-07-30
>
> We thank the reviewer for the thoughtful and constructive feedback. We address each point below.
>
> **Weaknesses:**
>
> In response, we have added two new synthetic high-dimensional examples (see the response to reviewer 1 (6kyE) for details) illustrating the utility of reduction mappings and their associated preconditioning, with a detailed analysis of the iteration complexity for each method. Moreover, our framework applies directly to prior work such as [46], where the authors employed a nonlinear reduction mapping—derived from an argmin formulation—to optimise towards the nearest ETF in the neural collapse setting. Their results, on both synthetic and real-world data, demonstrate accelerated convergence compared to standard training, further validating the benefits of this approach.
> Batch normalisation can also be formulated as a reduction mapping. However, due to its reliance on batch statistics, the more interesting and practical mappings are inherently stochastic. The analysis of such stochastic reduction mappings is an interesting direction and is left for future work. In contrast, a particularly instructive and tractable case is weight normalisation (WN). Consider a single neuron with weight vector $w \in \mathbb{R}^d$. WN adopts a scale–direction parametrisation:
> $$
> w = \Phi(\theta) := g \frac{v}{\|v\|}, \quad \theta = (v, g) \in \mathbb{R}^d \times \mathbb{R}.
> $$
> This defines a (chart) representation of the sphere bundle $S^{d-1} \times \mathbb{R} \to \mathbb{R}^d$. In the reduction view, the unit-norm direction serves as a fixed gauge, and the known scale–direction structure is captured via $\Phi$. Performing steepest descent in the pullback metric induced by $\Phi$ yields exactly the preconditioned update recommended by our framework.
> Letting $n = v/\|v\|$, the pullback metric is
> $$
> R = \mathrm{diag}\left( \frac{g^2}{\|v\|^2}(I - nn^\top), 1 \right).
> $$
> Because $R$ is singular (due to redundancy in $v$), the correct preconditioner is its Moore–Penrose pseudoinverse,
> $$
> R^\dagger = \mathrm{diag}\left( \frac{\|v\|^2}{g^2}(I - nn^\top), 1 \right),
> $$
> or alternatively, in a minimal coordinate chart, the full inverse is
> $$
> R^{-1} = \mathrm{diag}\left( g^{-2} I_{d-1}, 1 \right).
> $$
> In both cases, the inverse is closed-form and computationally efficient. At the layer level, the Jacobian becomes block-diagonal (one block per row), so preconditioning reduces to a per-row projection onto the tangent space followed by a scalar rescaling. The full metric remains block-diagonal with a cost of $\mathcal{O}(md)$ for a layer with $m$ neurons, which is negligible in practice.
> Future work will address the stochastic extension of reduction mappings and explore more real-world applications. Our theory suggests that while normalisation layers are already known to improve training speed in standard Euclidean settings, performing geometry-aware preconditioning through the pullback metric can yield further improvements in convergence—without additional wall-time overhead—since the metric inverses are inexpensive to compute.
>
> *Clarity:* Thank you for the suggestion. Due to space constraints, we couldn’t fit the whole motivation in the main text. In the final version of the paper, we will provide additional motivation in the main text.
>
> **Questions:**
>
> **1)** For the case of batch normalisation bounds, our results are tight in the sense that we can move between the two results in a straightforward manner. Specifically, when the batch normalisation transformation is interpreted as a reduction mapping $\Phi_{\text{BN}}$​, the pullback metric becomes a projector–rescale operator, and our framework recovers the same structural improvements in smoothness and gradient norms observed in prior BN analyses [44]. The multiplicative factor $\gamma^2/\sigma^2$ and the projection onto the orthogonal complement of the mean and normalised input directions emerge naturally from the geometry of the mapping and match the scaling and projection terms in the BN gradients.
> The difference lies in how the second-order effects are handled. In our general setting, when $\Phi$ is nonlinear, the curvature correction term involving $D^2\Phi$ is upper-bounded generically by the abstract quantity $QZ/m(\Phi)$, which captures the interaction between the curvature of the mapping and the gradient of the loss. In contrast, the BN analysis computes these same curvature terms explicitly using the algebraic structure of the batch statistics, leading to closed-form expressions for the full Jacobian and its effect on the gradient and Hessian. In this sense, the BN bounds can be seen as a concrete instance of our general result, where the curvature term is evaluated exactly rather than bounded.
> Therefore, up to this difference in treatment of curvature, the two sets of results are equivalent in spirit and mechanism. Our bounds can be viewed as a generalisation of the BN case, and when applied to the BN setting, they yield bounds that are tight up to the curvature correction. In particular, when the reduction mapping is affine—as in frozen BN—the curvature term vanishes, and our bounds match exactly. For nonlinear cases (e.g., batch-dependent BN), the tightness of our bound depends on how well the curvature correction is controlled, but the mechanism of improvement remains the same. For a general setting, our results are minimax-tight.
>
>
> **2)** The curvature and spectral gap conditions we impose are sufficient but not necessary for achieving improved smoothness and sharpness constants. In particular, they are not required to guarantee a qualitative improvement — that is, an improvement may still occur in their absence. In fact, the necessary and sufficient condition for a strict reduction in curvature appears in the Courant–Fisher formulation used in Lemma 4: the improvement occurs if and only if the feasible tangent space avoids the top eigenspace of the Hessian.
> What the curvature and spectral gap conditions provide is a quantitative strengthening of this principle. They turn the abstract variational condition into a concrete, verifiable, and practically useful bound. In that sense, our assumptions are best viewed as sufficient conditions for controlled and predictable improvement, which can guide algorithmic design and offer explicit guarantees in real-world applications.

---

> > ### Comment · Reviewer_rmzJ · 2025-08-05
> >
> > Great, thank you for the response. It answered all of my questions very satisfactorily.

---

### Official Review · Reviewer_6kyE · 2025-07-03

**Clarity:** 3
**Significance:** 2
**Originality:** 3
**Rating:** 4
**Confidence:** 3

**Summary:**

This paper presents a formal and general framework for understanding and quantifying the benefits of using "reduction mappings" in nonconvex optimization problems where the set of minimizers forms a manifold. The core idea is that by reparametrizing the problem to stay on a manifold that encodes known optimal structure, one can provably improve the local geometry of the objective function, leading to faster convergence for gradient-based methods.

**Questions:**

1. On the Practicality and Origin of the Reduction Mapping Ψ.

Could the paper provide a concrete, illustrative example from the literature cited (e.g., neural collapse [14], dictionary learning [22], or phase retrieval [31]) and explicitly formulate the corresponding reduction mapping Ψ? For instance:
In the context of neural collapse, where optimal classifiers collapse to an Equiangular Tight Frame (ETF), is Ψ a projection onto the ETF cone?
In low-rank matrix factorization, min ||X - UVᵀ||², where solutions are only unique up to an invertible transformation (UR, V(R⁻¹)ᵀ), how would one define Ψ to fix this ambiguity?

2. On the Verifiability of the Key Geometric Assumptions (ε and Δ)

Could the paper provide more intuition on the plausibility of these assumptions in real-world machine learning settings? Could you further connect your assumption of a spectral gap to the empirical findings on Hessian spectra in deep learning [66-70], which you cite? Does the "few large outlier eigenvalues" phenomenon directly support the existence of a Δmax > 0?

3. Lack of Experimental Results.

How large are the "spectral gaps" (Δmax, Δmin) in practice? How does the performance degrade as the assumptions (e.g., non-alignment) are gradually violated? It would help to answer these practical questions if experimental analysis could be involved.

**Ethical Concerns:**

["NO or VERY MINOR ethics concerns only"]

**Final Justification:**

The rebuttal addresses the concerns and questions well. The score is raised. Solid work.

**Limitations:**

yes

**Quality:**

2

**Strengths And Weaknesses:**

The main strength is the move from qualitative intuition to quantitative results. The paper provides explicit bounds on the new smoothness and sharpness constants. This requires non-trivial mathematical machinery, likely involving matrix analysis and differential geometry (as hinted by the use of pullback metrics).

The framework is powerful because it's general. It applies to both affine and nonlinear reductions and covers a wide range of practical scenarios mentioned in the introduction (neural collapse, matrix factorization, phase retrieval). It provides a principled explanation for why techniques like fixing parts of a model or using certain projections can accelerate training.

Weaknesses:

While the theoretical contribution is strong, there are areas that could be discussed for a complete picture, which is typical for a focused research paper.

1. Purely Theoretical: The paper is entirely theoretical and lacks experimental validation. While this is acceptable for a theory paper, even a simple numerical experiment on a synthetic problem (like the one in Figure 1) would make the theoretical gains more tangible.

2. Practicality of the Mapping Ψ: The framework assumes the optimal structure, and thus the reduction mapping Ψ, is known. In many real-world problems, this structure might only be known approximately, or it might emerge during training. The analysis doesn't account for errors in the mapping Ψ.

3. Computational Overhead: The proposed algorithm, Preconditioned Gradient Descent (PrecGD), requires computing and inverting the matrix R = DΦᵀDΦ. The paper acknowledges this cost but states that designing efficient solvers is "beyond the scope of this work." This is a fair statement, but it is the key barrier to making this framework immediately practical for complex, nonlinear mappings.

---

> ### Author Rebuttal · Authors · 2025-07-30
>
> We thank the reviewer for their thoughtful and constructive feedback. We address each point below.
>
> **Weaknesses:**
>
> **1)** We agree that numerical experiments can enhance intuition for the theory. In response, we conducted synthetic experiments illustrating the predicted iteration complexity improvements. These build upon Example 1 (Appendix A) and will appear in the final version with visualisations and analysis. All experiments were repeated with varied seeds/initialisations and results averaged.
> - **Example 1: Quadratic Toy Problem (Figure 1)**
>   Canonical function:
>   $$f(x_1,x_2) = x_1^2 + 10(x_2 − x_1)^2$$
>   with reduction mapping $x_2^\star(x_1) = x_1$, yielding reduced problem $F(x_1)=x_1^2$. GD is used with maximal step sizes.
> - **Example 2: High-Dimensional Quadratic**
>   Ill-conditioned problem ($K \in \mathbb{R}^{d\times d}, d=40$):
>   $$G(x, y) = \tfrac{1}{2}||x||^2 + \tfrac{1}{2}||y||^2 + \tfrac{\lambda}{2} ||y - Kx||^2$$
>   Reduction via $y = Kx$ gives:
>   $$F(x) = \tfrac{1}{2} x^\top (I + K^\top K) x,\quad R = I + K^\top K$$
> - **Example 3: Nonlinear Mapping (Tanh)**
>   Reduction $\Phi(x) = [x; \alpha \tanh(Kx)]$, full objective:
>   $$G(x, y) = \tfrac{1}{2}||x||^2 + \tfrac{1}{2}||y||^2 + \tfrac{\lambda}{2} ||y - \alpha \tanh(Kx)||^2$$
>   Reduced:
>   $$F(x) = \tfrac{1}{2} ||x||^2 + \tfrac{1}{2} \alpha^2 ||\tanh(Kx)||^2$$
>   with pullback metric:
>   $$R(x) = I + \alpha^2 K^\top \mathrm{diag}(\mathrm{sech}^4(Kx)) K$$
>
> |  Iterations to Convergence         | Full Problem  (GD) | Reduced (Euclidean GD) | Reduced (Pullback + PrecGD)                                  |
> |:---------:|:------------------:|:----------------------:|:------------------------------------------------------------:|
> | Example 1 | 800                | 1                      | 1                                                            |
> | Example 2 | 132,000            | 9,000                  | 1                                                            |
> | Example 3 | > 2,000,000        | 28,000                 | 20 (+1400 CG steps) |
>
> These results confirm that reduction significantly reduces iteration counts and preconditioning further enhances this—even for high-dimensional or nonlinear problems.
>
>
> **2)** While our theory assumes exact reduction mappings $\Psi$ for clarity, many practical cases admit explicit forms. Examples include BatchNorm, WeightNorm, or projections onto structured manifolds (sphere, Stiefel), often implemented via QR/polar decompositions. These are exact (not iterative) and fit our framework.
> Mappings derived as a solution of inner optimisation problems (e.g., balanced factorisations or ETF projections) are indeed approximate. We acknowledged this in Section 4.1, and handling such mappings is a key future direction.
>
>
> **3)** Inverting the pullback metric $R = D\Phi^\top D\Phi$ adds some overhead, but is tractable in many settings:
>
> - **Linear $\Phi$**: $R$ is constant with closed-form inverse or low-rank structure; cost is negligible.
> - **Structured nonlinear $\Phi$**: $R(x)$ often has favourable algebraic form (e.g. diagonal dominance, low-rank perturbation of identity). Efficient inversion via Woodbury or matrix identities is possible.
> - **Stable conditioning**: Well-behaved $\Phi$ (such as nonlinear projection on manifolds) keeps $R(x)$ stable, enabling iterative solvers (e.g. conjugate gradient) with only a few steps, hence they are not substantially increasing the wall-time performance as we see in Example 3.
>
> Thus, while efficient large-scale implementation is indeed an important direction beyond the current scope, we see no fundamental barrier to scalability in many settings. We have updated the discussion in the paper to better reflect these considerations.
>
> ---
>
> **Questions:**
>
>
> **1)** *Neural Collapse*: In the context of Neural Collapse, the reduction mapping $\Psi$ can be viewed as fixing classifier weights (equivalent mappings can be formulated for hidden layers via Deep NC characterisations) to a canonical or random ETF form, eliminating orthogonal symmetry. This constant mapping fits our framework.
> A richer formulation defines $\Psi$ implicitly via the nearest-ETF projection, as in [46]:
> $$U^\star \in \arg\min_{U\in St(d,C)} ||\bar{H} - UM||^2_F + \tfrac{\delta}{2} ||U - U_{\text{prox}}||^2_F$$
> This defines $\Psi(\bar{H}) = U^\star M$. Empirical validation in [46, Figs. 3,4] shows faster convergence and improved conditioning, matching our theoretical predictions.
>
>
> *Low-rank Factorisation*:
> Here, the solution is non-unique due to $\mathrm{GL}(r)$ symmetry: $(U, V) \mapsto (UR, V(R^{-\top}))$. Quotient manifold methods handle this via equivalence classes, but do not favour specific representatives.
>
> Our framework allows explicit representative selection—fixing a gauge—which is crucial when structure or regularity is needed (e.g., in generalisation).
>
> For example, an interesting choice is balanced factorisation: $U^\top U = V^\top V$. Let $A:=U^\top U, B:=V^\top V$. For any feasible $(U,V)$, compute the matrix geometric mean
> $$X: = A^{-1/2}(A^{1/2}BA^{1/2})A^{-1/2}$$
> and choose the principal symmetric PD square root:
> $$R(U, V) := X(U,V)^{1/2}$$
> so that $U^\top U = V^\top V$. This defines a reduction $\Psi(U,V) = (UR(U,V), V(R(U,V)^{-\top}))$.
> Unlike quotient optimisation, which simply eliminates symmetry, our mapping selects a preferred/specific representative from the equivalence class with desirable properties. Balancedness improves stability and generalisation in tasks like matrix completion and low-rank deep models.
>
> With this framework, we give practitioners the ability to act with principled control over the solution structure, enabling them to design problem-specific mappings that not only resolve symmetries but also accelerate convergence toward desirable and application-relevant solutions. More broadly, many important applications motivate the need for structure-aware optimisation. In this paper, we have chosen to focus on developing the general theoretical framework. In upcoming work, we will provide a detailed treatment of key applications—illustrating how reduction mappings and the associated geometry can be leveraged in practical settings across machine learning and general nonconvex problems.
>
> **2)** Τhe assumptions made in our framework are both plausible and robust in real-world machine learning settings. For the spectral gap $\Delta_{\max}$​, the assumption is satisfied in any anisotropic Hessian, which is virtually always the case in modern deep networks. The only exception is the isotropic case (all eigenvalues equal), which is highly non-generic and mostly of theoretical interest. In this degenerate scenario, our reduction framework does not improve convergence over classical training—precisely as our theory predicts. More generally, as shown in Section E.4, symmetric matrices with non-simple spectra form a measure-zero subset, and our analysis explicitly accommodates eigenvalue multiplicities in the top eigenspace as long as there is a gap below.
> For the non-alignment assumption $\varepsilon > 0$, geometrically this requires that the tangent space of the reduced manifold does not contain the top curvature directions we aim to eliminate. Such near-perfect alignment is also non-generic and would require adversarially structured mappings or objectives. Moreover, even if alignment does occur, small, structure-preserving perturbations to the reduction mapping (e.g., reparameterisations, gauge fixes) are sufficient to break it—without altering the semantics of the model or predictor.
> Regarding the connection to deep learning Hessian spectra, we note that the goal is not merely to argue that $\Delta_{\max} > 0$—this is almost always true in nontrivial problems—but to emphasise that the magnitude of the spectral gap in practice can be very large, as shown empirically in [66–70]. This has two key implications: (i) our theoretical gains via reduction and preconditioning become more pronounced in such settings, and (ii) even in the purely Euclidean reduced case, the large $\Delta_{\max}$ leads to a strictly smaller smoothness constant, and hence faster convergence. This explains the accelerated behaviour observed in real-world examples like neural collapse and ETF-aligned features [46], even without preconditioning.
>
> **3)** In practice, spectral gaps $⁡\Delta_{\min}$ and $\Delta_{\max}$​ are typically positive and well-separated, as confirmed in our experiments and supported by prior work on deep learning Hessians. These studies consistently observe a bulk–spike structure, where a few large outlier eigenvalues are separated from the bulk of smaller ones. This directly implies a large $\Delta_{\max}$, while $\Delta_{\min}$, though smaller, remains non-negligible. Such structure provides a natural setting where our assumptions hold and the benefits of reduction and preconditioning are most pronounced. For completeness, we include the spectral gaps for the three synthetic examples from above.
>
> - Example 1: $\Delta_{\max} = \Delta_{\min} = 40$ (2D case)
>
> - Example 2: $\Delta_{\max} = 3540, \Delta_{\min} = 20$
> - Example 3: $\Delta_{\max} = 3.465 \times 10^5, \Delta_{\min} = 1.646 \times 10^{-6}$

---

> > ### Comment · Reviewer_6kyE · 2025-08-05
> >
> > The rebuttal addresses the concerns and questions. Thus, the score is raised. Thanks.

---

### Decision · Program_Chairs · 2025-09-17

**Decision:**

Accept (spotlight)

**Comment:**

This is a strong and well-written paper that introduces a reduction-based framework for analyzing convergence rates of first-order methods in nonconvex optimization. The core idea is that by reparametrizing the problem to stay on a manifold that encodes known optimal structure, one can provably improve the local geometry of the objective function, leading to faster convergence for gradient-based methods. The framework proposed is quite general and appears to apply to many interesting scenarios in nonconvex optimization. It seems that it could have a significant impact on how to view many strategies incorporated in modern optimization methods. All the concerns raised during the reviewer discussions were adequately responded to in the rebuttal. This paper makes a valuable contribution and should be of broad interest to the NeurIPS community.